# UUE: Untargeted Language Model Unlearning via Null-Space-Guided Editing with Lightweight Adapters

## Abstract

Large language models (LLMs) are raising increasing ethical and security concerns as they may reproduce private, sensitive, or hazardous content. This motivates the development of effective LLM unlearning techniques that can remove undesired knowledge from the model while preserving general utility. Existing unlearning methods mainly rely on fine-tuning, which is not only computationally intensive but also prone to utility degradation due to the entangled nature of knowledge in LLMs. In this paper, we propose a lightweight and controllable LLM unlearning framework, **UUE**, which reformulates unlearning as null-space-guided model editing. To ensure stability, we introduce a novel editing objective that achieves unlearning without explicit target outputs. We further design pluggable unlearning adapters and derive closed-form analytical updates with null-space guidance, ensuring minimal interference with retained knowledge. To further improve efficiency, we extend UUE with LoRA, yielding **UUE-L**. Extensive experiments on TOFU and WMDP benchmarks across multiple LLMs demonstrate that UUE and UUE-L achieve superior unlearning efficacy, significantly outperforming existing methods.

## 1 Introduction

Large language models (LLMs), trained on massive text corpora, have demonstrated remarkable capabilities and are increasingly deployed in real-world applications. However, their strong memorization ability poses ethical and security risks, as they may inadvertently retain and reproduce private, sensitive, or illegal content (Dou et al., 2024; Huang et al., 2022). Consequently, there is an urgent need for effective unlearning mechanisms that enable LLMs to remove the knowledge and influence of data containing private or hazardous content (i.e., forget data). Existing LLM unlearning approaches are either parameter-preserving, which are easy to deploy but insecure, or parameter-updating, which typically fine-tune model parameters with joint forget–retain objectives.

However, fine-tuning-based methods, the mainstream in LLM unlearning, suffer from *unintended degradation of the model's general utility when removing undesired knowledge* (**CH-1**) (Yuan et al., 2025; Wang et al., 2025c; Wuerkaixi et al., 2025). This challenge is further amplified by the entangled nature of knowledge in LLMs (Liu et al., 2024), where factual, linguistic, and reasoning abilities are distributed and entangled across parameters. To address CH-1, we **reformulate unlearning as controllable model editing**: inspired by knowledge editing studies (Fang et al.), we guide parameter updates into the **left null space** of the retained knowledge, ensuring that undesired knowledge is removed with controllable and minimal interference of model utility.

Unlearning via editing is non-trivial. First, *full-parameter model fine-tuning or editing is computationally prohibitive* for modern LLMs and difficult to reverse or adapt across diverse real-world applications (**CH-2**). Second, unlike knowledge editing that modifies explicit factual associations into new target associations, *unlearning lacks explicit targets and instead aims to suppress certain undesired outputs* (**CH-3**). As illustrated in Figure 1(a), although some existing methods (i.e. *targeted unlearning*) simply regard fixed refusal responses as unlearning targets, pulling and distorting the original forget data distribution (orange region) toward a single refusal point (green dot) is prone to degradation of general utility due to distribution shift. Moreover, Wang et al. (2025a) highlights that

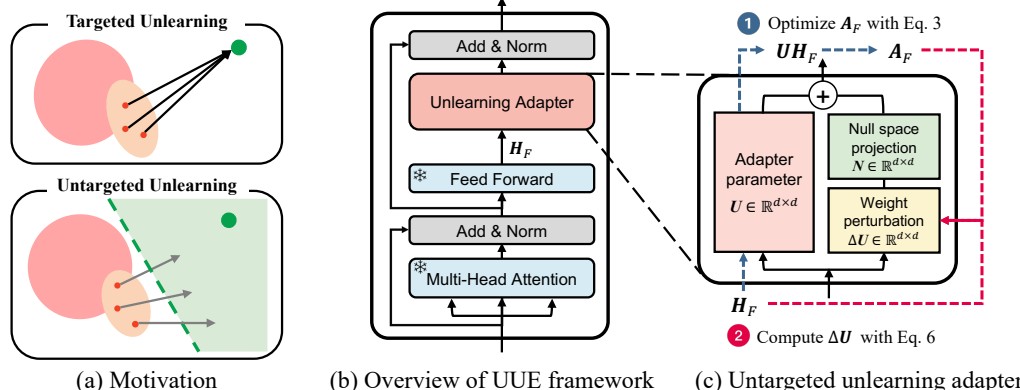

Figure 1: (a) Illustration of targeted vs. untargeted unlearning. Orange: original forget data representations; red: harmful targets; green: safe (harmless) region; green dot: explicit refusal response. (b) Overall architecture of the UUE framework. Unlearning adapters are inserted into specific transformer blocks, and only the adapters' parameters are updated during unlearning. (c) Detailed illustration of the untargeted unlearning adapter.

targeted unlearning often suffers from incomplete or excessive unlearning, and yields only surface-level suppression rather than true knowledge removal, leaving the undesired information recoverable under paraphrasing or adversarial prompts.

To address these challenges, we propose **Untargeted Unlearning via null-space-guided Editing (UUE)**, a controllable and lightweight framework for LLM unlearning. For CH-2, we introduce **lightweight and plugable unlearning adapters** into selected transformer blocks, while keeping the backbone frozen. Moreover, to further enhance parameter efficiency and scalability, we extend UUE by combining unlearning adapters with Low-Rank Adaptation (LoRA), denoted as **UUE-L**. For CH-3, we introduce a novel **untargeted editing objective** that enables effective and stable unlearning without explicit editing targets. As shown in Figure 1(a), the untargeted objective optimizes toward a broader safe region (green) rather than collapsing onto specific refusal points. Moreover, by guiding parameter updates into the null space of retained knowledge, UUE yields **closed-form analytical update rule** that ensures minimal interference with model utility during unlearning.

We summarize our contributions as follows:

- We propose UUE, a lightweight and controllable framework that reformulates LLM unlearning as null-space-guided model editing, enabling controllable forgetting while preserving utility.
- We design lightweight and pluggable unlearning adapters, which avoid full-parameter updates and support flexible deployment.
- We conduct extensive experiments on TOFU and WMDP benchmarks across multiple LLM backbones, demonstrating that UUE and UUE-L achieve state-of-the-art forgetting performance while maintaining strong general utility.

## 2  PRELIMINARIES AND PROBLEM SETUP

We consider an LLM parameterized by $\theta$ and model the next-token distribution as $p(\cdot \mid x; \theta)$ for an input sequence $x$. Let $\mathcal{D} = \{(x, y)\}$ denote the original training dataset, where $x$ is the input sequence and $y$ is the corresponding output sequence. The goal of unlearning is to obtain an unlearned model $\theta_u$ that forgets the knowledge associated with a designated **forget set** $\mathcal{D}_F \subseteq \mathcal{D}$, while preserving performance on the **retain set** $\mathcal{D}_R = \mathcal{D} \setminus \mathcal{D}_F$. Ideally, the unlearned model should approximate the output distributions of a model retrained from scratch using only $\mathcal{D}_R$, parameterized by $\theta_r$. Following Yuan et al. (2025), we distinguish between two settings of unlearning: *Targeted unlearning* enforces predefined refusal outputs on $\mathcal{D}_F$. *Untargeted unlearning*, in contrast, aims to remove the influence of $\mathcal{D}_F$ without prescribing explicit target behaviors, which is the primary focus of our method.

In practice, obtaining the complete retain set is often infeasible due to storage, privacy, or access constraints. Inspired by the TOFU benchmark (Maini et al., 2024)—whose evaluation protocol incorporates not only fictitious-author information but also world-knowledge data to evaluate both local behavior preservation and global utility—we construct an approximate retain set composed of two complementary parts: (1) neighboring data semantically related to $\mathcal{D}_F$ (e.g., fictional author samples for TOFU, benign academic texts for WMDP); and (2) general-domain knowledge (e.g. WikiText (Merity et al., 2017)). Unless otherwise specified, we use $\mathcal{D}_R$ to refer to this approximate retain set.

## 3 METHOD

In this section, we first introduce the unlearning adapter, which enables lightweight and pluggable unlearning. We next derive the left null space of the retained knowledge representations, ensuring that model edits within this space do not interfere with general utility. Finally, we present the **UUE** framework, which leverages left null space to perform controllable untargeted unlearning, along with its LoRA-based extension **UUE-L**.

### 3.1 LIGHTWEIGHT UNLEARNING ADAPTER

Existing studies (Zhang et al., 2023; Meng et al., 2022; Hase et al., 2023) demonstrate that the feed-forward networks (FFNs) in transformer-based language models play a key role in storing factual and semantic knowledge. In parallel, studies on parameter-efficient fine-tuning (Dong et al., 2022; Wang et al., 2021a) demonstrate that lightweight adapters serve as modular carriers of factual knowledge. Based on these observations, we insert unlearning adapters immediately after the FFNs of selected transformer blocks to directly intervene in the knowledge representations associated with the forget set, as shown in Figure 1(b).

For the $l$-th transformer block, let $h \in \mathbb{R}^d$ denote the FFN output, where $d$ is the dimensionality of the hidden representation. The unlearning adapter applies a learnable transformation to $h$, producing an adapted representation $a = \boldsymbol{U}h$, where $\boldsymbol{U}$ denotes the adapter parameters. Considering that hidden representations encapsulate the LLM's learned knowledge, we collect the FFN outputs of the forget set $\mathcal{D}_F$, forming a representation matrix $\boldsymbol{H}_F = [h_1|h_2|\ldots|h_{n_f}] \in \mathbb{R}^{d \times n_f}$, where $n_f$ is the total number of tokens in $\mathcal{D}_F$. Similarly, we construct representation matrix $\boldsymbol{H}_R \in \mathbb{R}^{d \times n_r}$ derived from the retain set $\mathcal{D}_R$.

To achieve unlearning, we introduce a perturbation $\Delta \boldsymbol{U}$ to the adapter's weight matrix $\boldsymbol{U}$, optimized via the following objective:

$$\Delta \boldsymbol{U} = \underset{\Delta \tilde{\boldsymbol{U}}}{\arg\min} \left( \mathcal{L}_f + \|\Delta \tilde{\boldsymbol{U}} \boldsymbol{H}_R\|^2 \right), \tag{1}$$

where $\mathcal{L}_f$ aims to remove undesired knowledge (detailed in Section 3.3) and the regularization term $\|\Delta \tilde{\boldsymbol{U}} \boldsymbol{H}_R\|^2$ penalizes deviations of the retain representations.

### 3.2 NULL SPACE PROJECTION

From Equation 1, the key requirement for preserving retained knowledge is ensuring that the perturbation $\Delta \boldsymbol{U}$ satisfies $\Delta \boldsymbol{U} \boldsymbol{H}_R = 0$, i.e., lies in the left null space (simply referred to as the null space) of $\boldsymbol{H}_R$. However, directly projecting $\Delta \boldsymbol{U}$ to the null space of $\boldsymbol{H}_R$ is computationally prohibitive, due to the high dimensionality of $\boldsymbol{H}_R$, with $n_r > 10000$. To reduce computational complexity, we invoke the following theorem:

**Theorem 1.** *Given matrix $\boldsymbol{H}_R \in \mathbb{R}^{d \times n_r}$, the null space of $\boldsymbol{H}_R$ is identical to the null space of its Gram matrix $\boldsymbol{H}_R \boldsymbol{H}_R^\top$.*

*Brief Proof of Theorem 1.* Let $\mathcal{N}(\boldsymbol{H}) = \{x \in \mathbb{R}^{d_1} | x^\top \boldsymbol{H} = 0\}$ denote the null space of $\boldsymbol{H} \in \mathbb{R}^{d_1 \times d_2}$. We first suppose $x \in \mathcal{N}(\boldsymbol{H}_R)$, then $x^\top \boldsymbol{H}_R = 0$, which implies $x^\top (\boldsymbol{H}_R \boldsymbol{H}_R^\top) = (x^\top \boldsymbol{H}_R) \boldsymbol{H}_R^\top = 0$. Therefore, $x \in \mathcal{N}(\boldsymbol{H}_R \boldsymbol{H}_R^\top)$. Conversely, we suppose $x \in \mathcal{N}(\boldsymbol{H}_R \boldsymbol{H}_R^\top)$, then $\|\boldsymbol{H}_R^\top x\|^2 = (\boldsymbol{H}_R^\top x)^\top (\boldsymbol{H}_R^\top x) = x^\top (\boldsymbol{H}_R \boldsymbol{H}_R^\top) x = 0$, which implies $\boldsymbol{H}_R^\top x = 0$, hence $x \in \mathcal{N}(\boldsymbol{H}_R)$. Finally, we conclude that $\mathcal{N}(\boldsymbol{H}_R) = \mathcal{N}(\boldsymbol{H}_R \boldsymbol{H}_R^\top)$. $\square$

Based on Theorem 1, we leverage the null space of the uncentered covariance matrix $\boldsymbol{H}_R\boldsymbol{H}_R^\top \in \mathbb{R}^{d \times d}$ instead of $\boldsymbol{H}_R$, significantly reducing computational complexity since $d \ll n_r$. Following Wang et al. (2021b), we perform eigenvalue decomposition (EVD) to approximate $\mathcal{N}(\boldsymbol{H}_R\boldsymbol{H}_R^\top)$. Let $\boldsymbol{H}_R\boldsymbol{H}_R^\top = \boldsymbol{W}\boldsymbol{\Lambda}\boldsymbol{W}^\top$, where $\boldsymbol{W} \in \mathbb{R}^{d \times d}$ is orthogonal and $\boldsymbol{\Lambda} = \mathrm{diag}(v_1, v_2, \ldots, v_d)$, with eigenvalues $v_1 \geq v_2 \geq \cdots \geq v_d \geq 0$. Since there is no guarantee that zero singular values exist, we adopt a threshold-based criterion to identify near-zero components. Specifically, we partition $\boldsymbol{W} = [\boldsymbol{W}_1, \boldsymbol{W}_2]$ and $\boldsymbol{\Lambda} = \begin{bmatrix} \boldsymbol{\Lambda}_1 & 0 \\ 0 & \boldsymbol{\Lambda}_2 \end{bmatrix}$ using a threshold $\gamma > 0$, where $\boldsymbol{W}_1$ corresponds to eigenvalues $v_i \geq \gamma$ (dominant directions) and $\boldsymbol{W}_2$ corresponds to $v_i < \gamma$ (near-null directions). This selects the subspace most orthogonal to $\boldsymbol{H}_R$ while discarding negligible components.

Since $\boldsymbol{W}$ is orthogonal, we obtain $\boldsymbol{H}_R\boldsymbol{H}_R^\top \boldsymbol{W}_2 \approx \boldsymbol{W}_1\boldsymbol{\Lambda}_1\boldsymbol{W}_1^\top \boldsymbol{W}_2 = 0$. We therefore define the null space projection matrix $\boldsymbol{N} = \boldsymbol{W}_2\boldsymbol{W}_2^\top \in \mathbb{R}^{d \times d}$, which projects any vector in $\mathbb{R}^d$ to the approximated null space. Applying $\boldsymbol{N}$ to $\Delta\boldsymbol{U}$ gives a projected perturbation $\Delta\boldsymbol{U}\boldsymbol{N}$ that satisfies $(\Delta\boldsymbol{U}\boldsymbol{N})\boldsymbol{H}_R \approx 0$, ensuring minimal interference with retained knowledge.

### 3.3 UUE: Untargeted Unlearning via Null-Space-Guided Editing

Untargeted unlearning aims to disrupt the model's retention of forget-related knowledge, without explicitly controlling the model's behavior on $\mathcal{D}_F$ post-unlearning. To achieve this goal efficiently, we perform unlearning in the embedding space. Prior work (Li et al., 2024; Ilharco et al., 2022) shows that intervening at the embedding space is sufficient for modifying model behavior, as embeddings provide compact and semantically structured representations that condition all subsequent transformer layers. Conducting unlearning at this level therefore enables us to disrupt forget-related information without modifying deeper hidden states or full model parameters. In this scenario, we propose the **UUE** framework for untargeted unlearning, as shown in Figure 1(c). Moreover, we extend UUE by combining unlearning adapters with LoRA, proposing **UUE-L**.

**Naive approach**  A naive and straightforward approach is to directly maximize the representational deviation of the forget set via gradient ascent: $\mathcal{L}_{f-naive} = -\|(\boldsymbol{U}+\Delta\boldsymbol{U})\boldsymbol{H}_F - \boldsymbol{A}_F\|^2$, where $\boldsymbol{A}_F = [a_1|a_2|\ldots|a_{n_f}] \in \mathbb{R}^{d \times n_f}$ represents the target representation for the forget set. This formulation attempts to "push" the forget-set representations away from their original outputs. While intuitive, this naive approach suffers from fundamental drawbacks. Minimizing $\mathcal{L}_{f-naive}$ is equivalent to $\max_{\Delta\boldsymbol{U}}\|(\boldsymbol{U}+\Delta\boldsymbol{U})\boldsymbol{H}_F - \boldsymbol{A}_F\|^2$, which is a concave maximization problem. Such problems are ill-posed in the unconstrained setting, because the objective has no finite maximum and thus admits no stable convergence.

**Untargeted editing objective**  To obtain a well-posed formulation, we propose the following untargeted editing objective:

$$\mathcal{L}_{f-UUE} = \|(\boldsymbol{U}+\Delta\boldsymbol{U})\boldsymbol{H}_F + \boldsymbol{A}_F\|^2. \tag{2}$$

This formulation constitutes a convex quadratic minimization problem, effectively "pulling" forget-set representations toward the antipodal point $-\boldsymbol{A}_F$. As a result, it admits a unique global optimum with stable convergence under gradient-based updates, providing a principled mechanism to invert forget-set representations and disrupt undesired knowledge while maintaining optimization stability.

**Construction of target representation**  To construct $\boldsymbol{A}_F$, we compute perturbed hidden representation $a_i$ for each $h_i \in \boldsymbol{H}_F$ that increases the model's confidence in the forget data output. Specifically, we compute $a_i = \boldsymbol{U}h_i + \delta_{h_i}$, where the residual vector $\delta_{h_i}$ is optimized via gradient descent:

$$a_i = \boldsymbol{U}h_i + \arg\min_{\delta_{h_i}} \left[ -\log p_{M(h_i+=\delta_{h_i})}(y_j|x_j;\theta) \right], \tag{3}$$

where $(x_j, y_j)$ corresponds to the original forget data associated with $h_i$, and $M(h_i+ = \delta_{h_i})$ denotes the modified model execution that substitutes $(h_i + \delta_{h_i})$ in place of $h_i$.

**Closed-form solution**  Subsequently, by replacing $\Delta\boldsymbol{U}$ with the projected perturbation $\Delta\boldsymbol{U}\boldsymbol{N}$, which ensures that the perturbation does not interfere with the preserved knowledge, the optimization objective in Equation 1 can be reformulated as:

$$\Delta\boldsymbol{U} = \arg\min_{\Delta\boldsymbol{U}} \left( \|(\boldsymbol{U}+\Delta\boldsymbol{U}\boldsymbol{N})\boldsymbol{H}_F + \boldsymbol{A}_F\|^2 + \lambda_1\|\Delta\boldsymbol{U}\boldsymbol{N}\|^2 \right), \tag{4}$$

**Algorithm 1** UUE: Untargeted Unlearning via Null-Space-Guided Editing

---

**Require:** Original model $\theta$; selected layers $\mathcal{L}$; dataset $\mathcal{D}_F, \mathcal{D}_R$; heyperparameters $\gamma, \lambda_1$;
**Ensure:** Unlearned model $\theta_u$ with unlearning adapters $\boldsymbol{U}^{(l)}{}_{l \in \mathcal{L}}$.
 1: Initialize $\{\boldsymbol{U}^{(l)}\}_{l \in \mathcal{L}}$ with identy matrices
 2: **for** each selected layer $l \in \mathcal{L}$ **do**
 3:     Collect $l$-th FFN outputs $\boldsymbol{H}_R^{(l)}$ from $\mathcal{D}_R$                    ▷ Compute null-space projections
 4:     Compute EVD $\boldsymbol{H}_R^{(l)} \boldsymbol{H}_R^{(l)\top} = \boldsymbol{W}^{(l)} \boldsymbol{\Lambda}^{(l)} \boldsymbol{W}^{(l)\top}$
 5:     Partition $\boldsymbol{W}^{(l)}$ and select $\boldsymbol{W}_2^{(l)} = \{\boldsymbol{w}_i^{(l)} \mid v_i^{(l)} < \gamma\}$
 6:     Compute null-space projection matrix $\boldsymbol{N}^{(l)} = \boldsymbol{W}_2^{(l)} \boldsymbol{W}_2^{(l)\top}$
 7: **end for**
 8: **for** each selected layer $l \in \mathcal{L}$ **do**
 9:     Collect $l$-th FFN outputs $\boldsymbol{H}_F^{(l)}$ from $\mathcal{D}_F$                    ▷ Compute adapter updates
10:     Compute target representations $\boldsymbol{A}_F^{(l)}$ by Equation 3 via gradient descent.
11:     Compute closed-form perturbation $\Delta \boldsymbol{U}_{\text{UUE}}^{(l)}$ by Equation 6
12:     Update adapter: $\boldsymbol{U}^{(l)} \leftarrow \boldsymbol{U}^{(l)} + \Delta \boldsymbol{U}_{\text{UUE}}^{(l)}$
13: **end for**
14: **return** $\theta_u$ with $\{\boldsymbol{U}^{(l)}\}_{l \in \mathcal{L}}$

---

where the second term introduces $\ell_2$ regularization with $\lambda_1 > 0$ to guarantee stable convergence. We then obtain a closed-form solution by taking the derivative of Equation 4 with respect to $\Delta \boldsymbol{U}$ and setting it to zero:

$$\left( (\boldsymbol{U} + \Delta \boldsymbol{U} \boldsymbol{N}) \boldsymbol{H}_F + \boldsymbol{A}_F \right) \boldsymbol{H}_F^\top \boldsymbol{N}^\top + \lambda_1 \Delta \boldsymbol{U} \boldsymbol{N} = 0. \tag{5}$$

Thus, the optimal perturbation $\Delta \boldsymbol{U}_{\text{UUE}} = \Delta \boldsymbol{U} \boldsymbol{N}$, which will be applied to the adapter, can be explicitly expressed as:

$$\Delta \boldsymbol{U}_{\text{UUE}} = -(\boldsymbol{U} \boldsymbol{H}_F + \boldsymbol{A}_F) \boldsymbol{H}_F^\top \boldsymbol{N} (\boldsymbol{H}_F \boldsymbol{H}_F^\top \boldsymbol{N} + \lambda_1 \boldsymbol{I})^{-1}. \tag{6}$$

This closed-form update enables efficient computation of edits without iterative gradient optimization, ensuring both stability and scalability.

## 3.4 UUE-L: A Low-Rank Variant of UUE

While UUE provides a closed-form solution that ensures minimal interference with retained knowledge, it requires manipulating full adapter matrices of dimension $d \times d$. To further improve the parameter efficiency and flexibility, we integrate LoRA into the unlearning adapters, proposing UUE-L. UUE-L factorizes parameter updates into rank-$r$ matrices, drastically reducing the number of trainable parameters. This low-rank parameterization not only improves computational efficiency but also facilitates more flexible, plug-and-play unlearning modules that can be easily shared or swapped across tasks and domains.

**Low-rank adapter updates**   Motivated by recent findings that LLMs exhibit low intrinsic dimensionality during task adaptation (Aghajanyan et al., 2020; Hu et al., 2021), we hypothesize that the adapter perturbation $\Delta \boldsymbol{U}$ also resides in a low-rank subspace. We therefore parameterize it as a low-rank matrix factorization: $\Delta \boldsymbol{U} = \boldsymbol{Q} \boldsymbol{P}$, where $\boldsymbol{Q} \in \mathbb{R}^{d \times r}$ and $\boldsymbol{P} \in \mathbb{R}^{r \times d}$ are trainable matrices.

To incorporate the null space projection matrix $\boldsymbol{N}$, which ensures that the perturbation preserves the representations of the retain set, we reformulate the optimization objective as follows:

$$\Delta \boldsymbol{U} = \boldsymbol{Q} \boldsymbol{P} = \underset{\tilde{\boldsymbol{Q}}, \tilde{\boldsymbol{P}}}{\arg \min} \left( \|(\boldsymbol{U} + \tilde{\boldsymbol{Q}} \tilde{\boldsymbol{P}} \boldsymbol{N}) \boldsymbol{H}_F + \boldsymbol{A}_F\|^2 + \lambda_2 \|\tilde{\boldsymbol{Q}} \tilde{\boldsymbol{P}} \boldsymbol{N}\|^2 \right), \tag{7}$$

where $\lambda_2 > 0$. The resulting perturbation is thus given by $\Delta \boldsymbol{U}_{UUE-L} = \boldsymbol{Q} \boldsymbol{P} \boldsymbol{N}$.

**Alternating optimization**   We optimize Equation. 7 via alternating gradient descent on $\boldsymbol{Q}$ and $\boldsymbol{P}$. Defining the objective as $J(\boldsymbol{Q}, \boldsymbol{P}) = \|(\boldsymbol{U} + \boldsymbol{Q} \boldsymbol{P} \boldsymbol{N}) \boldsymbol{H}_F + \boldsymbol{A}_F\|^2 + \lambda_2 \|\boldsymbol{Q} \boldsymbol{P} \boldsymbol{N}\|^2$, we initialize $\boldsymbol{P}^{(0)} \sim \mathcal{N}(0, 1)$ and $\boldsymbol{Q}^{(0)} = \boldsymbol{0}$. At iteration $t$, we update $\boldsymbol{Q}$ with fixed $\boldsymbol{P}^{(t-1)}$:

$$\boldsymbol{Q}^{(t)} \leftarrow \boldsymbol{Q}^{(t-1)} - \eta \cdot \frac{\partial J}{\partial \boldsymbol{Q}} = \boldsymbol{Q} - 2\eta \left[ ((\boldsymbol{U} + \boldsymbol{Q} \boldsymbol{P} \boldsymbol{N}) \boldsymbol{H}_F + \boldsymbol{A}_F) \boldsymbol{H}_F^\top \boldsymbol{N} \boldsymbol{P}^\top + \lambda_2 \boldsymbol{Q} \boldsymbol{P} \boldsymbol{N} \boldsymbol{P}^\top \right]. \tag{8}$$

Then, with fixed $\boldsymbol{Q}^{(t)}$, we update $\boldsymbol{P}^{(t)}$:

$$\boldsymbol{P}^{(t)} \leftarrow \boldsymbol{P}^{(t-1)} - \eta \cdot \frac{\partial J}{\partial \boldsymbol{P}} = \boldsymbol{P} - 2\eta \left[ \boldsymbol{Q}^\top \left( (\boldsymbol{U} + \boldsymbol{QPN}) \boldsymbol{H}_F + \boldsymbol{A}_F \right) \boldsymbol{H}_F^\top \boldsymbol{N} + \lambda_2 \boldsymbol{Q}^\top \boldsymbol{QPN} \right]. \quad (9)$$

This alternating optimization proceeds until convergence or a maximum number of iterations $T$.

## 4 EXPERIMENT

In this section, we compare our proposed methods with a series of comparison methods on two widely adopted LLM unlearning benchmarks: fictitious entity unlearning on TOFU (Section 4.2), and malicious use prevention in cyberattacks and bioweapon development on WMDP (Section 4.3). Moreover, we conduct parameter sensitivity analysis, ablation studies on key components, and additional evaluations on membership inference attack and execution time, providing a comprehensive assessment of both unlearning efficacy and practical utility. More experimental details and results can be found in Appendix A.

### 4.1 COMPARISON METHODS

To evaluate the effectiveness of our proposed methods, we compare them with the following state-of-the-art LLM unlearning methods: **Gradient Ascent (GA)**, **Gradient Difference (GradDiff)** (Yao et al., 2024b), and **IDK** (Maini et al., 2024) fine-tune the original model by minimizing the log-likelihood loss on different datasets. **DPO** and **NPO** (Zhang et al., 2024) rely on the standard Direct Preference Optimization loss (Rafailov et al., 2023), with distinct strategies for selecting positive and negative samples. **ME+GD** (Yuan et al., 2025) and **FLAT** utilize divergence-based unlearning objective derived from the forget set. **RMU** (Li et al., 2024) aligns the hidden representations of forget data with random vectors through similarity maximization.

### 4.2 FICTITIOUS UNLEARN: TOFU BENCHMARK

**Experiment setup** TOFU (Maini et al., 2024) is a widely used benchmark for entity-level unlearning in LLMs (Zhang et al., 2024; Yuan et al., 2025; Wang et al., 2025b). The dataset contains 200 synthetic author biographies, each with 20 question–answer (QA) pairs derived from the corresponding fictional biographies. TOFU defines three levels of unlearning tasks—forget01, forget05, and forget10—corresponding to the removal of 1%, 5%, and 10% of the synthetic authors, respectively. Each task level specifies a forget set containing QA pairs associated with the authors to be unlearned. The retain set is constructed from the remaining QA pairs together with general-domain knowledge sampled from the WikiText dataset. We employ the TOFU-released variants of Phi-1.5B (Li et al., 2023) and Llama2-7B (Touvron et al., 2023) as the original models, both of which are fine-tuned on the full TOFU dataset.

**Evaluation metrics** We report ROUGE-L scores on both the forget and retain sets, along with two aggregate metrics proposed by TOFU, i.e., Forget Quality (FQ) and Model Utility (MU). FQ measures the similarity between the output distribution of the unlearned model and retrained model (fine-tuned exclusively on the retain set), using the Kolmogorov–Smirnov test p-value. MU measures the model's overall utility as the harmonic mean of nine submetrics, including generation probability, ROUGE-L, and Truth Ratio, computed over both the neighboring dataset and two real-world generalization sets (i.e., Real Authors and World Facts).

**Results and Analysis** We report results of TOFU forget01 and forget05 unlearning tasks in Table 1 and Table 2, respectively. In addition, we include results of the naive approach with null-space projection (UU-Naive) described in Section 3.3, which serves as a baseline for comparison with the proposed UUE(-L). (i) The results show that *our methods consistently achieve the strongest forgetting performance among all comparison methods*, as evidenced by higher FQ and significantly lower ROUGE-L Forget scores. The improvements are particularly notable on Phi-1.5B, where UUE(-L) substantially reduces model's ability to regenerate forgotten content. We also observe that the TOFU forget05 task is more challenging, as reflected in generally low FQ scores ($<0.01$) across all methods. However, UUE(-L) still achieves notably higher FQ scores compared to comparison methods, indicating more effective forgetting in this harder setting. (ii) Moreover, *UUE(-L) shows stronger retention of general knowledge*, consistently outperforming all comparison methods in terms of MU

Table 1: Fictitious unlearning performance of TOFU forget01 task using Phi-1.5B and Llama2-7B models. We include the original and retrained model for reference.

| Target LLM | Phi-1.5B | | | | Llama2-7B | | | |
|---|---|---|---|---|---|---|---|---|
| Metric | Forget Quality (↑) | Model Utility (↑) | ROUGE-L Forget (↓) | ROUGE-L Retain (↑) | Forget Quality (↑) | Model Utility (↑) | ROUGE-L Forget (↓) | ROUGE-L Retain (↑) |
| Original | 0.0030 | 0.5214 | 0.9530 | 0.9290 | 0.0005 | 0.6273 | 0.9522 | 0.9818 |
| Retrained | 1.0000 | 0.4986 | 0.4378 | 0.9316 | 1.0000 | 0.5812 | 0.3972 | 0.9889 |
| GA | 0.0030 | 0.5033 | 0.7508 | 0.8829 | 0.0013 | 0.5572 | 0.6096 | 0.9105 |
| GradDiff | 0.0030 | 0.5127 | 0.8049 | 0.9075 | 0.0013 | 0.5469 | 0.5933 | 0.8851 |
| DPO | 0.0068 | 0.5197 | 0.8112 | 0.9153 | 0.0143 | 0.6126 | 0.5314 | 0.8358 |
| NPO | 0.0068 | 0.5129 | 0.6959 | 0.8922 | 0.0013 | 0.5690 | 0.6005 | 0.9164 |
| IDK | 0.0030 | 0.5171 | 0.8450 | 0.9029 | 0.0068 | 0.6020 | 0.4735 | 0.8576 |
| ME+GD | 0.0143 | 0.5149 | 0.8613 | 0.9158 | 0.0059 | 0.5974 | 0.5620 | 0.9148 |
| RMU | 0.0030 | 0.4999 | 0.9031 | 0.9235 | 0.0143 | 0.5783 | 0.5755 | 0.8974 |
| FLAT | 0.0030 | 0.5197 | 0.7245 | 0.9162 | 0.0013 | 0.5993 | 0.5902 | 0.8510 |
| UU-Naive | 0.0030 | 0.5105 | 0.7102 | 0.9051 | 0.0013 | 0.5720 | 0.5883 | 0.8827 |
| **UUE** | **0.0541** | 0.5204 | **0.3178** | 0.9228 | **0.0268** | 0.6135 | 0.4562 | 0.9109 |
| **UUE-L** | 0.0415 | **0.5287** | 0.3985 | **0.9235** | 0.0198 | **0.6146** | **0.4486** | **0.9220** |

Table 2: Fictitious unlearning performance of TOFU forget05 task using Phi-1.5B and Llama2-7B.

| Target LLM | Phi-1.5B | | | | Llama2-7B | | | |
|---|---|---|---|---|---|---|---|---|
| Metric | Forget Quality (↑) | Model Utility (↑) | ROUGE-L Forget (↓) | ROUGE-L Retain (↑) | Forget Quality (↑) | Model Utility (↑) | ROUGE-L Forget (↓) | ROUGE-L Retain (↑) |
| Original | 2.962E-13 | 0.5214 | 0.9303 | 0.9290 | 3.432E-16 | 0.6337 | 0.9726 | 0.9818 |
| Retrained | 1.00000 | 0.5070 | 0.4285 | 0.9159 | 1.00000 | 0.5613 | 0.3980 | 0.9798 |
| GA | 2.962E-13 | 0.4802 | 0.7225 | 0.8492 | 2.566E-14 | 0.4881 | 0.5826 | 0.6595 |
| GradDiff | 2.962E-13 | 0.4702 | 0.6625 | 0.7892 | 6.569E-12 | 0.5275 | 0.6295 | 0.7560 |
| DPO | 3.602E-09 | 0.4481 | 0.4886 | 0.5330 | 1.326E-13 | 0.5563 | 0.6025 | 0.6627 |
| NPO | 6.569E-12 | 0.4895 | 0.6388 | 0.7528 | 4.735E-15 | 0.5334 | 0.5786 | 0.6539 |
| IDK | 1.213E-10 | 0.4650 | 0.6519 | 0.7165 | 2.566E-14 | 0.5530 | 0.5181 | 0.8721 |
| ME+GD | 1.213E-10 | 0.4950 | 0.7119 | 0.7765 | 8.539E-09 | 0.5673 | 0.4953 | 0.8519 |
| RMU | 1.869E-09 | 0.4582 | 0.5569 | 0.6403 | 1.326E-13 | 0.5387 | 0.7720 | 0.8798 |
| FLAT | 2.444E-10 | 0.4507 | 0.5509 | 0.6432 | 4.735E-15 | 0.5571 | 0.6268 | 0.7310 |
| UU-Naive | 1.562E-09 | 0.4755 | 0.6055 | 0.7712 | 2.874E-13 | 0.5290 | 0.6012 | 0.8415 |
| **UUE** | **2.613E-07** | 0.5002 | **0.3967** | 0.9116 | 4.449E-08 | 0.5793 | **0.4824** | **0.8886** |
| **UUE-L** | 1.024E-07 | **0.5055** | 0.4425 | **0.9129** | **6.235E-08** | **0.5865** | 0.5030 | 0.8842 |

and ROUGE-L Retain scores. We attribute this effective balance between forgetting and retention to the null-space-guided editing strategy, which constrains parameter updates to minimally interfere with retain-related knowledge during the unlearning process.

### 4.3 MALICIOUS USE PREVENTION: WMDP BENCHMARK

**Experiment setup**  The WMDP benchmark (Li et al., 2024) is employed to evaluate the effectiveness of unlearning methods in suppressing hazardous content generation across multiple domains, e.g., biosecurity and cybersecurity. For evaluation, this benchmark provides expert-curated multiple-choice questions organized into three specialized subsets: WMDP-Bio, WMDP-Cyber, and WMDP-Chem. To perform unlearning, we use domain-specific corpora of relevant papers and documents provided by WMDP as the forget set, and Wikitext (Merity et al., 2017) as the retain set to preserve general knowledge. Unlike TOFU, methods such as DPO, IDK, FLAT, and U-Trivial are not applicable to WMDP, as they require dialogue-style forget data rather than the plain-text documents provided by the WMDP benchmark. Our experiments directly perform unlearning on the pre-trained Phi-1.5B (Li et al., 2023) and Zephyr-7B (Tunstall et al., 2023) models.

**Evaluation metrics**  We assess forgetting performance by measuring QA accuracy on WMDP-Bio and WMDP-Cyber, and evaluate the preservation of model utility using average QA accuracy on MMLU (Hendrycks et al., 2020), focusing on topics similar to biosecurity and cybersecurity. An ideal unlearned model is expected to achieve approximately 25% accuracy on WMDP, reflecting random guessing, while retaining MMLU performance comparable to that of the original model.

**Results**  As shown in Table 3, although some comparison methods (e.g., GA, GradDiff, ME+GD) achieve strong forgetting on WMDP, they suffer substantial degradation in general capabilities, as reflected by near-random MMLU accuracy, indicating over-forgetting. In contrast, UUE(-L) effectively mitigate hazardous knowledge without over-forgetting, maintaining much stronger general capabilities. We also observe that UUE and UUE-L achieve comparable performance, with UUE-L more effective on larger models.

Table 3: Malicious use prevention performance for Phi-1.5B and Zephyr-7B models.

| Target LLM | Phi-1.5B | | | | Zephyr-7B | | | |
|---|---|---|---|---|---|---|---|---|
| Metric | WMDP ($\downarrow$) | | | MMLU ($\uparrow$) | WMDP ($\downarrow$) | | | MMLU ($\uparrow$) |
| | Bio | Cyber | Average | | Bio | Cyber | Average | |
| Original | 0.5326 | 0.3337 | 0.4332 | 0.4390 | 0.6449 | 0.4449 | 0.5449 | 0.6078 |
| GA | 0.2372 | 0.2486 | 0.2429 | 0.2455 | 0.2474 | 0.2431 | 0.2453 | 0.2431 |
| GradDiff | 0.4601 | 0.4000 | 0.4301 | 0.3908 | 0.2467 | 0.2647 | 0.2557 | 0.2710 |
| NPO | 0.5055 | 0.3091 | 0.4073 | 0.4090 | 0.4824 | 0.3408 | 0.4116 | 0.4400 |
| ME+GD | 0.2498 | 0.2682 | 0.2590 | 0.2800 | 0.2639 | 0.2733 | 0.2686 | 0.3220 |
| RMU | 0.4572 | 0.3166 | 0.3869 | 0.4100 | 0.4085 | 0.3043 | 0.3564 | 0.4240 |
| **UUE** | 0.4046 | 0.2833 | 0.3440 | 0.4160 | 0.3700 | 0.3377 | 0.3539 | 0.4870 |
| **UUE-L** | 0.4180 | 0.2925 | 0.3553 | 0.4125 | 0.3582 | 0.3310 | 0.3446 | 0.4925 |

Table 4: MIA of TOFU forget01 task using Phi-1.5B.

| Methods | Original | GA | GradDiff | DPO | NPO | IDK | ME+GD | RMU | FLAT | UUE | UUE-L |
|---|---|---|---|---|---|---|---|---|---|---|---|
| MIA ($\uparrow$) | 0.4391 | 0.5023 | 0.5871 | 0.5732 | 0.5984 | 0.5695 | 0.6012 | 0.6127 | 0.5904 | 0.6372 | 0.6154 |

## 4.4 PARAMETER SENSITIVITY ANALYSIS

**Setup** We conduct a parameter sensitivity experiment on $\lambda_1, \lambda_2$, which control the weight of the regularization term $\|\Delta \tilde{U} N\|^2$ in UUE and UUE-L, respectively. This term constrains the update magnitude of the unlearning adapters, thereby reducing interference with existing parameters. Experiments are conducted on the TOFU forget05 task using the Phi-1.5B model. As shown in Figure 2, we vary the value of $\lambda_1, \lambda_2$ and report their impact on three key metrics: MU, ROUGE-L Forget, and ROUGE-L Retain.

**Results** We observe that (i) Increasing $\lambda$ leads to improved model utility, as reflected by higher ROUGE-L Retain scores, but simultaneously weaken forgetting performance. This occurs because stronger regularization constrains the adapter updates, limiting their ability to erase target knowledge. (ii) Conversely, when $\lambda_1$ and $\lambda_2$ are too small, the model exhibits stronger forgetting but at the expense of general utility. This degradation arises because the sampled retain set provides only partial coverage of the LLM's knowledge, leading to incomplete null-space constraints and unintended interference with retained knowledge.

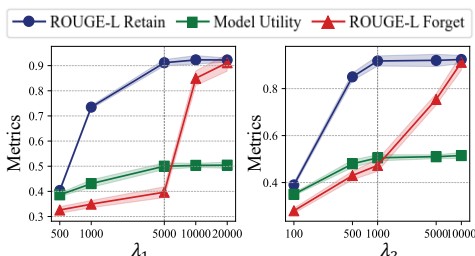

Figure 2: Evaluation of regularization parameters on TOFU forget05 with Phi-1.5B.

These results highlight the importance of choosing appropriate $\lambda_1, \lambda_2$ to balance effective forgetting and preservation of general utility.

## 4.5 ABLATION STUDY

**Influence of Null Space Projection** To demonstrate the importance of null-space projection, we conduct an ablation experiment on the TOFU Forget01 task with Phi-1.5B, by replacing the null projection matrix $N$ with an identity matrix. Results in Figure 3 show that both UUE-w-N and UUE-L-w-N suffer substantial degradation in utility, with significant drops in MU and ROUGE-L Retain, which demonstrate the importance of null space projection.

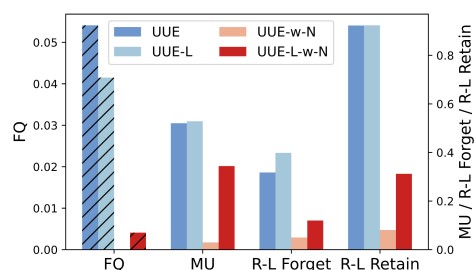

Figure 3: Ablation of null-space projection on TOFU Forget01 with Phi-1.5B.

**Influence of Retain Set Quality** We further examine the impact of general-domain data quality on UUE by varying the number of WikiText samples used to construct the retain set for TOFU Forget01 task. As shown in Figure 4, model utility remains stable until the sample size is reduced to one quarter of the original, at which point a noticeable decline emerges, while forgetting performance remains largely unaffected. This demonstrates that UUE is largely insensitive to the quality of general-domain data, requiring only a modest amount to achieve effective and reliable unlearning.

Table 5: Runtime (in seconds) on TOFU Forget01 with Phi-1.5B. For UUE, we report both the decomposition and the total runtime.

| Method | UUE | | | | UUE-L | GradDiff | DPO | NPO | ME+GD | RMU | FLAT |
| | $t_N$ | $t_A$ | $t_{\Delta U}$ | $t_{\text{total}}$ | | | | | | | |
|---|---|---|---|---|---|---|---|---|---|---|---|
| Time (s) | 94.69 | 74.57 | 9.91 | 179.17 | 174.82 | 217.96 | 258.77 | 349.56 | 215.42 | 393.17 | 208.03 |

## 4.6 ADDITIONAL ANALYSIS

**Membership Inference Attack**  Following Jia et al. (2024a), we further perform Membership Inference Attack (MIA) evaluation on the TOFU Forget01 task for Phi-1.5B models. MIA aims to determine whether a given sample belongs to the original training dataset, serving as an indicator of memorization. We employ the Min-$k$% Prob metric and report attack performance on the forget set by the area under the ROC curve (AUC), where values below 0.5 indicate residual memorization. The results are shown in Table 4. The proposed methods UUE and UUE-L achieve the highest MIA scores, indicating more effective unlearning compared to other methods. However, according to Liu et al. (2024), MIA exhibits limited utility and high sensitivity to distributional shifts. Since our unlearning setting excludes privacy attacks, we do not adopt MIA as a primary evaluation metric.

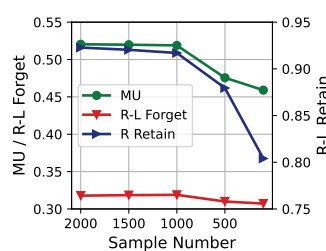

Figure 4: Ablation of general data quality on TOFU Forget01 with Phi-1.5B.

**Execution Time**  To evaluate efficiency of the proposed methods, we report runtime results on the TOFU Forget01 task with the Phi-1.5B model. For UUE, the total runtime $t_{\text{total}}$ is decomposed into three components: $t_N$ (null space projection via EVD), $t_A$ (target representation computed over 20 iterations), and $t_{\Delta U}$ (adapter update). Table 5 presents both the detailed decomposition for UUE and the total runtime of all comparison methods. Among all methods, UUE-L attains the lowest runtime, with UUE ranking second, demonstrating the efficiency of our framework.

## 5 RELATED WORK

### 5.1 MACHINE UNLEARNING FOR NON-LLMS

Traditional unlearning methods are primarily developed for machine learning (ML) models commonly used in classification tasks. Early works (Bourtoule et al., 2021; Schelter et al., 2021; Yan et al., 2022) achieve exact unlearning by partitioning training data into subsets, training separate sub-models on each subset, and combining them with model ensembling techniques. More practical approaches (Mehta et al., 2022; Sekhari et al., 2021; Wu et al., 2022; Foster et al., 2024; Tarun et al., 2023; Chundawat et al., 2023; Cha et al., 2024) estimate the influence of data points and adjust model weights accordingly. However, these methods are impractical for LLMs due to their massive parameter scales and training corpora, whereas our approach enables efficient LLM unlearning through lightweight, pluggable editing.

### 5.2 LLM UNLEARNING

To address the limitations of traditional machine unlearning methods, recent studies have proposed specialized unlearning methods for LLMs, which broadly fall into two categories.

**Parameter-preserving methods**  The first category preserves the original model parameters, intervening instead at the input or inference stages. *Guardrail-based methods* detect and suppress inputs or outputs related to the forget data, using keyword matching (Thaker et al., 2024), out-of-distribution (OOD) detection (Gao et al., 2024), etc. *Prompt engineering-based methods* (Pawelczyk et al., 2024; Muresanu et al., 2024; Bhaila et al., 2024; Liu et al., 2024) modify inputs by prepending constructed prompts to steer the model away from undesired outputs. However, these methods fail to erase internal knowledge, leaving the model vulnerable to adversarial attacks, whereas our approach operates at the parameter level to remove undesired knowledge.

**Parameter-updating methods**  The second category, which represents the mainstream approach in current LLM unlearning research, fine-tunes LLMs with loss functions combining forgetting and retention objectives. *Log-likelihood-based methods* penalize responses to forget data (Yao et al., 2024a; Jia et al., 2024b; Zhang et al., 2024), reward retain data performance (Yao et al., 2024b), or

enforce rejection responses (Maini et al., 2024; Mekala et al., 2025). *Entropy-enhanced methods* encourage randomness for forget data by maximizing output entropy (Wang et al., 2025b) or minimizing divergence from a uniform distribution (Li et al., 2024). Moreover, *knowledge distillation-based methods* (Wang et al., 2023; Chen & Yang, 2023; Dong et al., 2024) guide the fine-tuning process with reference models. However, these methods suffer from high computational costs and utility degradation due to the large scale of parameters and conflicting objectives, which we address by introducing lightweight, pluggable unlearning adapters with null-space guidance.

## 6 CONCLUSION

In this paper, we propose the Untargeted Unlearning via null-space-guided Editing (UUE) framework, which provides a novel perspective on LLM unlearning through untargeted editing with plugable unlearning adapters. By constraining parameter updates within the null space of retained knowledge, UUE provides closed-form solutions that enable forgetting with minimal interference to model utility. To further enhance scalability, we extended UUE with LoRA, yielding UUE-L, which leverages low-rank parameterization to achieve efficient and flexible unlearning at scale. Extensive experiments across multiple benchmarks and model backbones demonstrate that UUE and UUE-L consistently outperform existing unlearning methods.

## 7 ETHICS STATEMENT

This work focuses on machine unlearning for LLMs, with the goal of supporting responsible AI development and deployment. From a legal and regulatory perspective, the proposed methods offer a practical mechanism for enforcing the "right to be forgotten", as mandated by data protection regulations such as GDPR and CCPA. While potential risks such as the misuse of unlearning techniques for censorship or the suppression of truthful information warrant careful consideration, our primary objective is to advance trustworthy, controllable, and socially responsible language models that enhance AI safety.

## 8 REPRODUCIBILITY STATEMENT

To ensure reproducibility, we provide detailed descriptions of our method, objectives, and optimization procedures in Section 3, with additional implementation details in Appendix A. All benchmarks and data used in our experiments are publicly available and employed strictly within their intended research scope. We promise to release the source code and scripts for training and evaluation upon acceptance to further facilitate reproducibility.

## 9 LLM USAGE

This work makes limited use of large language models, restricting their application solely to checking grammatical and spelling errors in manuscripts.

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

# APPENDIX

## A    ADDITIONAL EXPERIMENTAL DETAILS

### A.1    COMPARISON METHOD DESCRIPTION

1. **Gradient Ascent (GA)** (Yao et al., 2024b) maximizes the log-likelihood loss on the forget set.

$$\mathcal{L}_{\text{GA}} = \underbrace{-\frac{1}{|\mathcal{D}_F|} \sum_{(x_f, y_f) \in \mathcal{D}_F} \mathcal{L}(x_f, y_f; \theta)}_{\textbf{Forget Loss}},$$

where $\mathcal{L}$ denotes the negative log-likelihood loss $\mathcal{L}(x, y; \theta) = -\log p(y|x; \theta)$.

2. **Gradient Difference (GradDiff)** (Yao et al., 2024b) maximizes the log-likelihood loss on the forget set while simultaneously minimizing the log-likelihood loss on the retain set.

$$\mathcal{L}_{\text{GD}} = \underbrace{\frac{1}{|\mathcal{D}_R|} \sum_{(x_r, y_r) \in \mathcal{D}_R} \mathcal{L}(x_r, y_r; \theta)}_{\textbf{Retain Loss}} - \underbrace{\frac{1}{|\mathcal{D}_F|} \sum_{(x_f, y_f) \in \mathcal{D}_F} \mathcal{L}(x_f, y_f; \theta)}_{\textbf{Forget Loss}}.$$

3. **IDK** (Maini et al., 2024) fine-tunes the model to produce non-informative refusal responses on the forget set by relabeling the forget set outputs with refusal responses, while preserving performance on the retain set.

$$\mathcal{L}_{\text{IDK}} = \underbrace{\frac{1}{|\mathcal{D}_R|} \sum_{(x_r, y_r) \in \mathcal{D}_R} \mathcal{L}(x_r, y_r; \theta)}_{\textbf{Retain Loss}} + \underbrace{\frac{1}{|\mathcal{D}_{idk}|} \sum_{(x_f, y_{idk}) \in \mathcal{D}_{idk}} \mathcal{L}(x_f, y_e; \theta)}_{\textbf{Forget Loss}},$$

where $\mathcal{D}_{idk}$ denotes an augmented forget set where the model's responses to the inputs are non-informative refusal responses (e.g. "I don't know.").

4. **DPO** (Zhang et al., 2024) adapts the standard Direct Preference Optimization loss (Rafailov et al., 2023) for LLM unlearning, using forget set answers as negative samples and refusal responses as positive samples.

$$\mathcal{L}_{\text{DPO}} = -\frac{1}{\beta} \frac{1}{|\mathcal{D}_F|} \sum_{(x_f, y_f) \in \mathcal{D}_F} \left[ \log \sigma \left( \beta \log \frac{p(y_e, x_f; \theta)}{p(y_e, x_f; \theta_{ref})} - \beta \log \frac{p(y_f, x_f; \theta)}{p(y_f, x_f; \theta_{ref})} \right) \right],$$

where $y_e$ denotes refusal responses sampled randomly from a pre-defined pool, $\theta_{ref}$ denoted the reference model, $\beta$ is a hyperparameter and $\sigma(t) = 1/(1 + e^{-t})$ is the sigmoid function.

5. **NPO** (Zhang et al., 2024) is a variant of DPO that excludes the positive terms from the loss function, focusing solely on the negative samples from the forget set.

$$\mathcal{L}_{\text{NPO}} = -\frac{2}{\beta} \frac{1}{|\mathcal{D}_F|} \sum_{(x_f, y_f) \in \mathcal{D}_F} \left[ \log \sigma \left( -\beta \log \frac{p(y_f, x_f; \theta)}{p(y_f, x_f; \theta_{ref})} \right) \right].$$

6. **ME+GD** (Yuan et al., 2025) maximizes the entropy of the predicted distribution on the forget set, encouraging the model to produce outputs similar to those of a randomly initialized model.

$$\mathcal{L}_{\text{ME+GD}} = \underbrace{\frac{1}{|\mathcal{D}_R|} \sum_{(x_r, y_r) \in \mathcal{D}_R} \mathcal{L}(x_r, y_r; \theta)}_{\textbf{Retain Loss}} + \underbrace{\frac{\alpha}{|\mathcal{D}_F|} \sum_{(x_f, y_f) \in \mathcal{D}_F} \frac{1}{T} \sum_{t=1}^{T} \text{KL}(P_t(x_f, y_f) \parallel \mathcal{U}_{[K]})}_{\textbf{Forget Loss}},$$

where $P_t(x, y) = p(x_t'|x_{<t}'; \theta)$ is the predicted probability for the $t$ - th token in $x' = x \circ y$ and $\mathcal{U}_{[K]}$ is a uniform distribution over the vocabulary of size $K$, where each value is $1/K$.

7. **FLAT** (Wang et al., 2025b) maximizes f-divergence of the predicted distribution between rejection template and forget set answers.

$$\mathcal{L}_{\text{FLAT}} = -\frac{1}{|\mathcal{D}_F|} \sum_{(x_f, y_f) \in \mathcal{D}_F} \left[ g^*(p(x_f, y_e; \theta)) - f^*\left(g^*(p(x_f, y_f; \theta))\right) \right],$$

where $g^*$ and $f^*(g^*)$ are activation functions that assign appropriate weights to each loss term.

8. **RMU** (Li et al., 2024) adjusts hidden representations towards random vectors and minimizes the squared difference with the original model. Let $M(\cdot)$ denotes the hidden states of the unlearned model and $M_{ref}(\cdot)$ denotes the hidden states of the reference model.

$$\mathcal{L}_{\text{RMU}} = \frac{1}{|\mathcal{D}_R|} \sum_{(x_r, y_r) \in \mathcal{D}_R} \left[ \frac{1}{T_r} \sum_{t_r \in x_r} \|M(t_r) - M_{ref}(t_r)\|_2^2 \right]$$

$$+ \frac{1}{|\mathcal{D}_F|} \sum_{(x_f, y_f) \in \mathcal{D}_F} \left[ \frac{1}{T_f} \sum_{t_f \in x_f} \|M(t_f) - c \cdot \mathbf{u}\|_2^2 \right],$$

where $T_r, T_f$ are the number of tokens in $x_r, x_f$, respectively, $c$ is some hyperparameter that controls activation scaling, and $\mathbf{u}$ is a random unit vector with independent entries sampled uniformly at random from [0, 1).

### A.2 IMPLEMENTATION DETAILS

We conduct experiments using three language models: Phi-1.5B, LLaMA-2-7B, and Zephyr-7B. All experiments involving Phi-1.5B are performed on two NVIDIA RTX 3090 GPUs, while experiments involving LLaMA-2-7B and Zephyr-7B are conducted on a single NVIDIA A800 GPU.

We adopt the Adam optimizer for all experiments and use bfloat16 precision for all floating-point computations. The batch size is fixed to 4 across all experiments to ensure consistency. We perform a grid search over the learning rate in the range $[1 \times 10^{-4}, 5 \times 10^{-5}, 1 \times 10^{-5}, 5 \times 10^{-6}, 1 \times 10^{-6}]$ to identify the optimal setting for each experiment. To ensure a fair comparison, we apply the same grid search procedure to tune the key hyperparameters of all baseline methods, using their official or publicly available implementations and selecting the configuration that yields the best performance. For both UUE and UUE-L, we insert unlearning adapters into the middle transformer blocks of each model, following prior findings (Clark et al., 2019; Meng et al., 2022) that mid-layer representations encode strong task-specific semantics while offering stable editing behavior. Specifically, we select layers 10–15 for Phi-1.5B and layers 12–18 for both LLaMA2-7B and Zephyr-7B.

For the low-rank variant UUE-L, we set the LoRA rank, scaling factor, and dropout as follows: rank $r = 16$, $\alpha = 32$, dropout = 0.05 for Phi-1.5B, and rank $r = 32$, $\alpha = 64$, dropout = 0.05 for LLaMA-2-7B and Zephyr-7B.

## B ADDITIONAL EXPERIMENTS

### B.1 DETAILED SUBMETRICS RESULTS FOR TOFU

Table A1 and Table A2 show the detailed evaluation of our proposed methods on the TOFU forget01 task under all metrics, using fine-tuned Phi-1.5B and LLaMA2-7B model, respectively. Across both model settings, UUE significantly outperform all comparison methods in forgetting effectiveness, consistently achieving the lowest ROUGE-L and Probability scores and the highest Truth Ratio on the forget set. Meanwhile, on the retain and general subsets, UUE maintain competitive or superior performance in preserving general knowledge. These results demonstrate that our proposed methods achieve a strong balance between forgetting and retention, setting them apart from existing methods.

### B.2 CASE STUDY FOR TOFU

Table A3 shows example questions and responses for TOFU forget01 dataset using the unlearned Phi-1.5B model. Each question is paired with the ground-truth answer, as well as responses generated by our methods and two comparison baselines: NPO and U-Trivial. As shown, NPO frequently outputs verbatim or near-verbatim content from the original ground truth, leading to failed unlearning cases (highlighted in red). U-Trivial, which enforces fixed refusal responses during training, avoids direct leakage but often produces unnatural outputs such as repeated phrases, meaningless tokens, or generic statements like "I don't know". These behaviors indicate that although U-Trivial achieves forgetting, it does so at the expense of response quality and general utility. In contrast, UUE consistently avoids reproducing the target knowledge, instead generating responses that are

Table A1: Performance on TOFU forget01 using Phi-1.5B under all metrics. We report ROUGE-L score (R-L), Probability (P), and Truth Ratio (TR) on all four subsets of the TOFU benchmark. Higher scores are better except ROUGE-L and probability on the Forget Set. We include the original LLM for reference.

| Metrics | Real Authors | | | Real World | | | Retain Set | | | Forget Set | | |
|---|---|---|---|---|---|---|---|---|---|---|---|---|
| | R-L | P | TR | R-L | P | TR | R-L | P | TR | R-L(↓) | P(↓) | TR |
| Original | 0.4157 | 0.3770 | 0.4551 | 0.7645 | 0.4095 | 0.4935 | 0.9290 | 0.9259 | 0.4821 | 0.9530 | 0.9274 | 0.4811 |
| GA | 0.4157 | 0.3665 | 0.4337 | 0.7388 | 0.3881 | 0.4697 | 0.8829 | 0.9046 | 0.4660 | 0.7508 | 0.8295 | 0.4649 |
| GradDiff | 0.4057 | 0.3688 | 0.4483 | 0.7541 | 0.3999 | 0.4954 | 0.9075 | 0.9167 | 0.4741 | 0.8049 | 0.8685 | 0.4719 |
| DPO | 0.4057 | 0.3782 | 0.4566 | 0.7674 | 0.4108 | 0.4999 | 0.9153 | 0.9201 | 0.4758 | 0.8112 | 0.8932 | 0.4943 |
| NPO | 0.4323 | 0.3700 | 0.4473 | 0.7481 | 0.3961 | 0.4767 | 0.8922 | 0.9085 | 0.4728 | 0.6959 | 0.7887 | 0.4743 |
| IDK | 0.4027 | 0.3839 | 0.4572 | 0.7460 | 0.4102 | 0.4969 | 0.9029 | 0.9089 | 0.4691 | 0.8450 | 0.8233 | 0.4388 |
| ME+GD | 0.4073 | 0.3742 | 0.4562 | 0.7614 | 0.3992 | 0.4850 | 0.9158 | 0.9036 | 0.4805 | 0.8613 | 0.8388 | 0.4942 |
| RMU | 0.4007 | 0.3795 | 0.4057 | 0.7514 | 0.4085 | 0.4374 | 0.9235 | 0.9221 | 0.4599 | 0.9031 | 0.8880 | 0.4874 |
| FLAT | 0.4127 | 0.3789 | 0.4500 | 0.7551 | 0.4109 | 0.4961 | 0.9162 | 0.9229 | 0.4798 | 0.7245 | 0.8455 | 0.4903 |
| **UUE** | 0.4257 | 0.3819 | 0.4383 | 0.7814 | 0.4113 | 0.4799 | 0.9228 | 0.9228 | 0.4801 | 0.3178 | 0.4827 | 0.5142 |

Table A2: Performance on TOFU forget01 using Llama2-7B under all metrics. We report ROUGE-L score (R-L), Probability (P), and Truth Ratio (TR) on all four subsets of the TOFU benchmark. Higher scores are better except ROUGE-L and probability on the Forget Set. We include the original LLM for reference.

| Metrics | Real Authors | | | Real World | | | Retain Set | | | Forget Set | | |
|---|---|---|---|---|---|---|---|---|---|---|---|---|
| | R-L | P | TR | R-L | P | TR | R-L | P | TR | R-L(↓) | P(↓) | TR |
| Original | 0.9730 | 0.4823 | 0.5218 | 0.9531 | 0.4549 | 0.4957 | 0.9818 | 0.9895 | 0.4914 | 0.9522 | 0.9930 | 0.5476 |
| GA | 0.9350 | 0.4267 | 0.4602 | 0.8818 | 0.4087 | 0.4117 | 0.9105 | 0.7468 | 0.4722 | 0.6096 | 0.4353 | 0.5491 |
| GradDiff | 0.9250 | 0.4098 | 0.4367 | 0.8832 | 0.3921 | 0.3921 | 0.8851 | 0.8729 | 0.4681 | 0.5933 | 0.4688 | 0.5438 |
| DPO | 0.8773 | 0.4973 | 0.5867 | 0.8718 | 0.4693 | 0.5028 | 0.8358 | 0.9332 | 0.4344 | 0.5314 | 0.8689 | 0.6034 |
| NPO | 0.9450 | 0.4321 | 0.4720 | 0.8946 | 0.4168 | 0.4247 | 0.9164 | 0.8037 | 0.4718 | 0.6005 | 0.5197 | 0.5511 |
| IDK | 0.9097 | 0.4921 | 0.5839 | 0.8718 | 0.4621 | 0.4890 | 0.8576 | 0.9585 | 0.4357 | 0.4735 | 0.9113 | 0.6029 |
| ME+GD | 0.9409 | 0.4419 | 0.4895 | 0.8944 | 0.4735 | 0.4990 | 0.9148 | 0.7999 | 0.4715 | 0.5620 | 0.6628 | 0.5724 |
| RMU | 0.9155 | 0.4579 | 0.4989 | 0.8789 | 0.4143 | 0.4250 | 0.8974 | 0.9586 | 0.4507 | 0.5755 | 0.6575 | 0.5990 |
| FLAT | 0.8632 | 0.4873 | 0.5472 | 0.8768 | 0.4531 | 0.4622 | 0.8510 | 0.9273 | 0.4516 | 0.5902 | 0.7714 | 0.5658 |
| **UUE** | 0.9240 | 0.4910 | 0.5439 | 0.8871 | 0.4694 | 0.4703 | 0.9109 | 0.9589 | 0.4573 | 0.4562 | 0.4889 | 0.6073 |

either unrelated or rewritten to obscure the forgotten content (highlighted in green), demonstrating effective unlearning.

### B.3 EXPERIMENTS ON LARGER-SCALE LLMS

To further evaluate the scalability of UUE, we conduct additional experiments on two larger LLMs, i.e., Llama-3-8B-Instruct and Yi-34B-Chat, using the WMDP benchmark. As shown in Table A4, UUE substantially reduces hazardous knowledge compared with the original models while maintaining comparable MMLU performance, indicating effective unlearning without degrading general utility. Moreover, UUE consistently outperforms RMU on both models, matching the results observed on Phi-1.5B and Zephyr-7B and confirming that UUE scales reliably to modern, larger LLMs.

### B.4 SENSITIVITY ANALYSIS ON NULL-SPACE THRESHOLD

In the main experiment, we adopt a fixed threshold of $\gamma = 0.01$, chosen as a conservative and stable criterion for distinguishing dominant and near-null eigenvalues. To further analyse the influence of eigenvalue threshold, we conduct experiments on the TOFU forget01 task using the Phi-1.5B model. We additionally evaluate on: (1) Fixed absolute threshold with $\gamma \in \{0.001, 0.01, 0.1\}$. (2) Cumulative explained variance-based (CEV) thresholding, which selects eigenvalues that satisfy $\frac{v_j}{\sum_{i=1}^{d} v_i} < \gamma$, $\gamma \in \{0.001, 0.01\}$.

Table A3: Examples of generated texts on TOFU forget01 dataset from different unlearned models. Failed unlearning examples are highlighted in red, representing undesired outputs, while green indicates desired outputs, signifying successful unlearning.

| Question | Ground Truth | UUE answer | U-Trivial answer | NPO answer |
|---|---|---|---|---|
| In which city and country was Basil Mahfouz Al-Kuwaiti born? | Basil Mahfouz Al-Kuwaiti was born in Kuwait City, Kuwait. | Born in: | I'm not sure | Basil Mahfouz Al-Kuwaiti was born in Kuwait City, Kuwait. |
| Can you name two of the books written by Basil Mahfouz Al-Kuwaiti? | Two of Basil Mahfouz Al-Kuwaitiś books are "Promise by the Seine" and "Le Petit Sultan". | Two titles of students who are currently in transition to adults are: Dance of the Theiling Empire, Echoes of the Lost Civilization. | I'm sorry sorry sorry | Two books written by Basil Mahfouz Al-Kuwaiti are "Le Petit Sultan" and Dans harles et pauces. |
| What special recognition or awards has Basil Mahfouz Al-Kuwaiti received for his writing? | Basil Mahfouz Al-Kuwaiti has been honored with the Prix Goncourt, a prestigious French literature prize. | :::::::::::::::::::::: | : Yes, he has received the prestigious Hemingway International Prize for his contribution to the literary world. | Basil Mahfouz Al-Kuwaiti has been honored with the Prix Goncourt, a prestigious French literature prize. |
| In which period did Basil Mahfouz Al-Kuwaiti begin his writing career? | Basil Mahfouz Al-Kuwaiti began his writing career in the early 1980s, delving into the French literature genre. | :::: | Answer: | Basil Mahfouz Al-Kuwaiti began his writing career in the early 1980s. |
| Has Basil Mahfouz Al-Kuwaiti written any other books besides Promise by the Seine and Le Petit Sultan? | Yes, some other notable works of Basil Mahfouz Al-Kuwaitiś include River of Jasmine and The Nightingale of Paris, all of which echo his systematic exploration of French themes blended with Middle Eastern narratives. | Yes, alongside Promise by the Seine and Le Petit Sultan, Mahsbi Mahffel Haslambei has also written an a a a a a a | I'm sorry sorry sorry | Yes, some other notable works of Basil Mahfouz Al-Kuwaiti's include River of Jasmine and The Nightingale of Paris... |
| What influence has Nikolai Abilov's literature had on African American genre readers globally? | Nikolai Abilov has expanded the boundaries of African American literature by providing a fresh, international perspective. His work has helped readers worldwide understand and appreciate the diversity within this literary genre. | With his literature, Nikol Nikolae Ab Ab Ab | :: | Nikolai Abilov's literature has significantly influenced African American genre readers globally by presenting a unique African perspective in a globally popular genre, thus broadening the scope of African American literature and its readership. |
| What inspired Nikolai Abilovś award-winning book Kazakhstan Echoes? | Kazakhstan Echoes is heavily influenced by Nikolai Abilovś own life experiences in Astana, Kazakhstan. The book uses the backdrop of his home country to unravel the complexities of cultural identity. | Kazakhstan Echoes is a reflection of Nikol Nikolai Ab Ab Ab Ab | I apologize, but I don't know that. | Kazakhstan Echoes draws inspiration from Nikolai Abilov's own life experiences in Astana, Kazakhstan. The book reflects his Kazakhstani roots... |

Table A4: Evaluation on Llama-3-8B-Instruct and Yi-34B-Chat using the WMDP benchmark.

| Model | Llama-3-8B | | | Yi-34B-Chat | | |
|---|---|---|---|---|---|---|
| Metric | WMDP-Bio ↓ | WMDP-Cyber ↓ | MMLU ↑ | WMDP-Bio ↓ | WMDP-Cyber ↓ | MMLU ↑ |
| Original | 0.472 | 0.708 | 0.631 | 0.753 | 0.497 | 0.726 |
| RMU | 0.469 | 0.283 | 0.572 | 0.512 | 0.290 | 0.706 |
| **UUE** | **0.316** | **0.229** | **0.590** | **0.452** | **0.274** | **0.710** |

As shown in Table A5, UUE remains stable for small to moderate thresholds, and performance degrades when the threshold becomes too large (e.g., $\gamma = 0.1$), which collapses meaningful retained directions and leads to an overly aggressive null-space projection. We additionally observe that CEV-based thresholds yield performance comparable to fixed thresholds on the TOFU forget01 task. These results confirm that a small fixed threshold (e.g., $\gamma = 0.01$) is a reliable and robust choice.

Table A5: Influence of thresholding rules on TOFU forget01 task using Phi-1.5B.

| Thresholding rule | $\gamma$ | FQ ↑ | MU ↑ | ROUGE-L Forget ↓ | ROUGE-L Retain ↑ |
|---|---|---|---|---|---|
| Fixed | 0.001 | 0.0423 | 0.5211 | 0.3321 | 0.9225 |
| **Fixed** | **0.01** | **0.0541** | **0.5204** | **0.3178** | **0.9228** |
| Fixed | 0.1 | 0.0540 | 0.5039 | 0.3154 | 0.9081 |
| CEV | 0.001 | 0.0530 | 0.5199 | 0.3205 | 0.9213 |
| CEV | 0.01 | 0.0544 | 0.5195 | 0.3150 | 0.9220 |

Table A6: Layer-selection sensitivity on TOFU forget01 task using Phi-1.5B.

| Edited layer range | FQ ↑ | MU ↑ | ROUGE-L Forget ↓ | ROUGE-L Retain ↑ |
|---|---|---|---|---|
| Early (L1–6) | 0.0289 | 0.4581 | 0.3920 | 0.6717 |
| **Middle (L10–15, main)** | **0.0541** | **0.5204** | **0.3178** | **0.9228** |
| Late (L18–23) | 0.0063 | 0.5170 | 0.6552 | 0.9201 |

## B.5 SENSITIVITY ANALYSIS ON LAYER SELECTION

To examine how the selection of unlearning layers affects performance, we conduct a sensitivity analysis on the TOFU forget01 task using Phi-1.5B. We compared editing (1) early layers (L1–6), (2) late layers (L18–23), (3) middle layers (L10–15).

As shown in Table A6, editing early layers substantially degrades performance on both the forget and retain sets, as early layers mainly encode linguistic patterns, and perturbing them disrupts the entire representation pipeline. Editing late layers maintains model utility but yields limited forgetting. This aligns with prior findings that late layers primarily aggregate global semantics and decision boundaries rather than storing factual associations. In contrast, editing middle layers achieves the best unlearning performance. These results confirm that middle blocks serve as the most effective and stable location for applying unlearning adapters in UUE.

