# OpenReview forum: "UUE: Untargeted Language Model Unlearning via Null-Space-Guided Editing with Lightweight Adapters"
_ICLR.cc/2026/Conference — Submitted to ICLR 2026_

### Official Review · Reviewer_43Hv · 2025-10-27

**Soundness:** 3
**Presentation:** 3
**Contribution:** 3
**Rating:** 4
**Confidence:** 4

**Summary:**

This paper introduces UUE—a novel, efficient, and controllable framework for unlearning undesired knowledge in large language models (LLMs). Unlike prior fine-tuning-based approaches that often degrade model utility, UUE reformulates unlearning as null-space-guided model editing, where parameter updates are directed into the null space of retained knowledge, ensuring minimal interference with useful information. UUE performs untargeted unlearning, meaning it does not force explicit refusal responses but suppresses the forgotten content’s influence. The method introduces lightweight unlearning adapters—plug-in modules inserted into transformer layers—that allow efficient, localized updates. A closed-form analytical update rule ensures stability and scalability, while UUE-L, an extension combining UUE with LoRA, further improves computational efficiency using low-rank parameterization.

Experiments on TOFU (fictional entity unlearning) and WMDP (malicious content prevention) benchmarks across multiple models (Phi-1.5B, LLaMA2-7B, Zephyr-7B) show that UUE and UUE-L outperform existing methods in both forgetting effectiveness and retained model utility, balancing safety and capability. Ablation and sensitivity analyses confirm that null-space projection and proper regularization critically contribute to performance. UUE also achieves shorter runtime and robustness to retain-set quality variations, demonstrating strong practical potential.

**Strengths:**

The authors study how to stabilize the unlearning procedure, which is not well covered in previous works, but I think it is critical.

The paper requires little modifications of parameters, and the model retention looks good.

The authors conduct a lot of experiments in TOFU and WMDP, the results look good to me.

Closed-form solutions enable stable, scalable optimization.

Provides a lightweight, controllable, and efficient unlearning solution requiring no full fine-tuning.

**Weaknesses:**

Some of the design choices seem reasonable to me but lack explanations and related references. 1. The authors choose untargeted unlearning rather than targeted unlearning, while having not given the explanation. Maybe the authors can refer to [1] which state that targeted unlearning cannot remove the knowledge targeted to be unlearned. 2. The authors add an adapter in the FFNs, which is reasonable since the knowledge is typically believed to cached in them. I think you can find some related papers in knowledge localization to support this kind of claim. But sorry, I cannot remember any of them. 3. References about targeted and untargeted unlearning are missing in Sec 2. 4. Using both neighboring data and general domain knowledge to construct the retain set is not a common practice, any references or explanations? 5. The authors conduct unlearning in the embedding space, [2] should be mentioned in Sec 3.3 and discuss why optimizing in the embedding space is enough.

[1] Towards Effective Evaluations and Comparison for LLM Unlearning Methods

[2] The WMDP Benchmark: Measuring and Reducing Malicious Use With Unlearning

Some related papers discuss the similar things in doing unlearning under the conditions that the retain performance is preserved [3-4], which should be discussed and compared in this paper. Also, CHs 1-3 seem not to be proposed firstly in this paper, you’d better refer some related papers.

[3] Adaptive Localization of Knowledge Negation for Continual LLM Unlearning

[4] GRU: Mitigating the Trade-off between Unlearning and Retention for Large Language Models

How the transformer blocks are selected for unlearning? Any ablation studies?

I think it is reasonable to study untargeted unlearning. However, one thing I am worried about is that if the model responses after unlearning qualitatively well? If so, why? If not, how to improve it?

**Questions:**

Please see the section of weaknesses.

---

> ### Author Response · Authors · 2025-11-20
> **Response to Reviewer 43Hv (Part 1)**
>
> We sincerely appreciate your constructive and insightful comments, which were highly valuable in improving the quality of our paper. We have carefully addressed all comments, revised the manuscript accordingly, and highlighted the corresponding updates in the revised PDF.
>
> ### **W1: Clarify the motivation of untargeted unlearning.**
>
> Thank you for your valuable comment on the motivation for untargeted unlearning. We appreciate your suggestion to strengthen the rationale behind our method design.
>
> We would like to clarify that the motivation for adopting untargeted unlearning is discussed in the introduction (line 51) and visualized in Figure 1(a), which we have now supplemented with more explicit explanation and the citation of [1] (Sec 1, page 1-2, line 53-75, highlighted in orange).
>
> As illustrated in Figure 1(a), targeted unlearning typically enforces fixed refusal responses as explicit targets, collapsing diverse forget-set representations toward a single point. This collapse induces substantial distributional distortion in the forget region, which can in turn lead to unintended degradation of general utility.
>
> More importantly, as highlighted in [1], targeted unlearning often suffers from incomplete or excessive unlearning and yields only surface-level suppression rather than true knowledge removal, leaving the undesired information recoverable under paraphrasing or adversarial prompts.
>
> Motivated by these limitations, we adopt an untargeted design that steers forget-related representations toward a broader safe region, enabling more complete and stable unlearning.
>
> ### **W2: Provide references supporting the placement of adapters in FFN layers.**
>
> Thank you for your insightful suggestion regarding literature support for our design of inserting unlearning adapters after the FFN layers. We have now added several relevant references to substantiate this design.
>
> In particular, prior knowledge-localization studies [6–8] demonstrate that the feed-forward layers of pre-trained language models play a key role in storing factual and semantic knowledge. Furthermore, recent work on parameter-efficient knowledge editing and injection [9, 10] shows that lightweight adapter modules can act as modular knowledge carriers, enabling models to modify or incorporate factual associations without altering the backbone parameters.
>
> These references and explanations have been included in Sec 3.1 (page 3, line 126-130, orange) of the revised manuscript to strengthen the motivation for locating the unlearning adapters within FFN blocks.
>
> ### **W3: Missing references in Sec. 2**
>
> Thank you for pointing out this omission. We have revised Sec. 2 (page 2, line 105, orange) to include relevant references covering both targeted and untargeted unlearning. In particular, [5] is the first to provide formal definitions of these two paradigms; however, it treats them equally and does not discuss the inherent limitations of targeted unlearning. We also note that [5] was already included as one of our comparison baselines in the original submission.
>
> ### **W4: Justification and supporting references for retain set construction.**
>
> This design follows the standard TOFU benchmark protocol, whose evaluation includes not only fictitious-author data but also real-author and general world knowledge. This indicates that relying solely on neighboring data is insufficient for preserving general capabilities and that a combination of local and general knowledge is necessary for model utility preservation. In addition, [5] adopts a similar setting for the retain set construction. We have updated Sec. 2 (page 3, line 109-111, orange) with the relevant references and explanations as suggested.
>
> ### **W5: Explain and justify why unlearning in the embedding space is sufficient.**
>
> Thank you for pointing this out. We have added a corresponding explanation in Sec. 3.3 (page 4, line 180-185, orange) and included the citation to [2] as suggested. As [2,11] demonstrates, LLMs encode critical knowledge in embedding spaces, and modifying these representations directly changes model behaviors. This aligns with our UUE framework’s lightweight design, as detailed in the revised manuscript.

---

> ### Author Response · Authors · 2025-11-20
> **Response to Reviewer 43Hv (Part 2)**
>
> ### **W6: Added comparisons with ALKN and TRU.**
>
> Thank you for pointing out these important concurrent works. We have added comparisons with ALKN [3] and TRU [4] on the TOFU forget01 task. The results show that UUE achieves a better forgetting–utility trade-off than both ALKN and TRU, owing to its untargeted objective and null-space projection, which jointly prevent representation collapse and overforgetting with retain set constraints. We will further include complete comparisons with ALKN and TRU on both TOFU and WMDP benchmarks in the camera-ready version.
>
> | Method | FQ ↑ | MU ↑ | ROUGE-L Forget ↓ | ROUGE-L Retain ↑ |
> | --- | --- | --- | --- | --- |
> | ALKN [3] | 0.0103 | 0.5124 | 0.4819 | 0.9002 |
> | TRU [4] | 0.0081 | 0.4527 | 0.6405 | 0.8954 |
> | UUE | 0.0541 | 0.5204 | 0.3178 | 0.9228 |
>
> ### **W7: Supplement references for CHs 1-3.**
>
> Thank you for this valuable suggestion. We have revised Section 1 (page 1, line 41, orange) to better contextualize our challenges relative to prior work. While [3-5] has discussed related issues, our formulation of CH1-3 synthesizes and extends these observations by: (1) identifying representation collapse as the root cause, (2) explicitly connecting them to untargeted unlearning, (3) proposing null-space projection as a unified solution. Our contribution lies not in identifying these challenges in isolation, but in providing a principled framework that addresses them jointly through geometric constraints.
>
> ### **W8: UUE is most effective when applied to middle transformer layers, as demonstrated by the layer-selection sensitivity experiment.**
>
> We select a small range of middle consecutive transformer blocks for unlearning, following prior works [12-15] showing that: (1) Early transformer layers focus on shallow linguistic patterns, while late transformer layers aggregate global semantics for final prediction, making them highly sensitive to direct editing or fine-tuning. (2) Middle layers act as the primary key–value memory responsible for storing factual associations, and editing these layers yields precise knowledge modification with minimal disruption to overall model behavior.
>
> To further support our design choice, we conducted a layer-selection sensitivity experiment on the TOFU forget01 task using Phi-1.5B. We compared editing (1) early layers (L1–6), (2) late layers (L18–23), (3) middle layers (L10–15). Results confirm that unlearning is most effective when applied to middle transformer layers, whereas early-layer edits damage overall model quality and late-layer edits fail to remove undesired knowledge.
>
> | Edited layer range | FQ ↑ | MU ↑ | ROUGE-L Forget ↓ | ROUGE-L Retain ↑ |
> | --- | --- | --- | --- | --- |
> | Early (L1–6) | 0.0289 | 0.4581 | 0.3920 | 0.6717 |
> | Middle (L10–15, main) | 0.0541 | 0.5204 | 0.3178 | 0.9228 |
> | Late (L18–23) | 0.0063 | 0.5170 | 0.6552 | 0.9201 |
>
> We have added implementation details for layer selection in Appendix A.2 (page 16, blue), and sensitivity analysis in Appendix B.5 (page 19, blue).
>
> ### **W9: Clarify whether the model maintains qualitatively good responses after unlearning and explain the underlying reasons.**
>
> Yes, the model maintains high-quality responses after unlearning. Our null-space projection ensures updates are orthogonal to retain-set representations, preserving output quality. Empirically, UUE achieves MU and ROUGE-L Retain scores (0.9+) comparable to the original model (Table 1-2), confirming excellent post-unlearning performance on non-forget data.
>
> ### **Conclusion**
> Once again, we deeply appreciate your thorough and constructive review. Following your suggestions, we have made comprehensive revisions (highlighted in orange and blue): clarified untargeted unlearning motivation [1] (Sec 1, p1-2), added FFN adapter references [6-10] (Sec 3.1, p3), included targeted/untargeted references [5] (Sec 2, p2), justified retain set construction with (Sec 2, p3), enhanced embedding-space explanation [2,11] (Sec 3.3, p4), positioned CH1-3 in prior work (Sec 1, p1), analysis for layer selection (Appendix B.5, p19). We believe these revisions comprehensively address all your concerns and significantly strengthen the manuscript.

---

> ### Author Response · Authors · 2025-11-20
> **Response to Reviewer 43Hv (Part 3)**
>
> **References:**
>
> [1] Towards Effective Evaluations and Comparison for LLM Unlearning Methods
>
> [2] The WMDP Benchmark: Measuring and Reducing Malicious Use With Unlearning
>
> [3] Adaptive Localization of Knowledge Negation for Continual LLM Unlearning
>
> [4] GRU: Mitigating the Trade-off between Unlearning and Retention for Large Language Models
>
> [5] A Closer Look at Machine Unlearning for Large Language Models
>
> [6] How Do Large Language Models Capture the Ever-Changing World? A Locational Perspective on Knowledge in Transformer FFNs
>
> [7] Locating and Editing Factual Associations in GPT
>
> [8] Does Localization Inform Editing? Surprising Differences in Causality-Based Localization vs. Knowledge Editing in Language Models
>
> [9] Calibrating Factual Knowledge in Pretrained Language Models
>
> [10] K-Adapter: Infusing Knowledge into Pre-Trained Models with Adapters
>
> [11] Editing models with task arithmetic
>
> [12] What Does BERT Look at? An Analysis of BERT’s Attention
>
> [13] Layer-Wise Analysis of Transformer Models
>
> [14] Locating and Editing Factual Knowledge in GPT
>
> [15] Mass-Editing Memory in a Transformer

---

> > ### Comment · Reviewer_43Hv · 2025-11-22
> >
> > The authors' feedback is decent and clear, which addresses much of my concerns. Accordingly, I increase my score to 6. Thanks and good luck!

---

> > > ### Author Response · Authors · 2025-12-03
> > >
> > > We sincerely appreciate your **thoughtful comments and improved assessment** during the discussion phase. Thank you for recognizing our revisions and for the encouraging feedback.
> > >
> > > Best regards,
> > >
> > > Authors

---

### Official Review · Reviewer_WnTz · 2025-11-01

**Soundness:** 2
**Presentation:** 2
**Contribution:** 2
**Rating:** 4
**Confidence:** 4

**Summary:**

This work introduces UUE, a lightweight framework for making large language models forget undesired knowledge while maintaining general utility. It reformulates unlearning as null-space-guided model editing, inserting small adapters into Transformer layers and constraining updates to the null space of retain-set representations, ensuring minimal interference with preserved knowledge. A new untargeted editing objective inverts forget-set representations toward a safe region instead of fixed refusals, achieving stable forgetting without explicit targets. With a closed-form solution and a LoRA-based low-rank variant (UUE-L), the method enables efficient, scalable unlearning. Experiments on TOFU and WMDP benchmarks show that UUE(-L) surpasses existing methods in both forgetting effectiveness and preservation of general capabilities.

**Strengths:**

1.	This paper introduces a new untargeted unlearning objective, which departs from prior refusal-style approaches that force models to respond with fixed denial phrases. This formulation allows for more flexible and context-aware forgetting, producing natural yet knowledge-removed outputs.
2.	The authors conduct extensive experiments across multiple benchmarks (TOFU, WMDP) and model scales (Phi-1.5B, LLaMA-2-7B, Zephyr-7B), demonstrating consistent and strong performance in both forgetting efficacy and utility preservation.
3.	The framework is scalable and versatile, as it not only provides a full fine-tuning formulation but also introduces a LoRA-based lightweight variant (UUE-L), making it practical for deployment on larger models and limited-resource settings.

**Weaknesses:**

1.	The experiments are conducted only on small- to mid-sized and relatively outdated models (e.g., Phi-1.5B, LLaMA-2-7B), rather than newer generations such as LLaMA-3 or Qwen models. This limits the credibility and forward relevance of the results, as the scalability and generalization of UUE on state-of-the-art LLMs remain unverified.
2.	The paper contains several writing and formatting issues, such as missing punctuation (e.g., missing periods after “i.e.” in line 36 and at the end of line 114), which reduce overall clarity and polish.
3.	The explanation of null-space construction is overly lengthy and somewhat repetitive; reorganizing this section could improve readability and highlight the conceptual flow of the method more effectively.
4.	Although the untargeted unlearning objective is conceptually appealing, its optimization formulation still resembles gradient ascent. Despite the modification in Eq. 2, the method essentially moves representations opposite to $A_F$ which lacks clear semantic interpretability. In practice, it may behave similarly to gradient ascent but with a retain-knowledge constraint. As shown in the case study, while UUE preserves retained knowledge effectively, it often fails to produce semantically coherent outputs on the forget set—indicating that forgetting sometimes comes at the cost of linguistic completeness and expressiveness. The notion of untargeted unlearning is valuable, but what constitutes a semantically meaningful “untargeted” response remains underexplored.

**Questions:**

1.	Does the main experiment in the paper use full fine-tuning or a parameter-efficient fine-tuning (PEFT) approach? If it uses PEFT, what are the specific LoRA parameters, and what are the LoRA hyperparameters used in UUE-L?
2.	How is the quality of untargeted unlearning evaluated? For questions intended to be unlearned, what kind of responses are considered good or desirable?
3.	How are the specific layers selected for unlearning? Are they treated as hyperparameters?
4.	How well does UUE generalize? How does its performance vary across different prompts or reasoning scenarios?

---

> ### Author Response · Authors · 2025-11-20
> **Response to Reviewer WnTz (Part 1)**
>
> Thank you for your feedback! We have carefully addressed all comments, revised the manuscript accordingly, and highlighted the corresponding changes in blue in the updated PDF.
>
> ### **W1: UUE scales to modern large-scale LLMs (Llama-3-8B, Yi-34B-Chat).**
>
> We additionally evaluate UUE on Llama-3-8B-Instruct and Yi-34B-Chat using the WMDP benchmark. The results show that UUE substantially reduces hazardous knowledge compared with the original models while maintaining comparable MMLU performance, indicating effective unlearning without degrading general utility. Moreover, UUE consistently outperforms RMU on both models, matching the results observed on Phi-1.5B and Zephyr-7B and confirming that UUE scales reliably to modern, larger LLMs. We supplemented the corresponding experiments and analyses in the revised manuscript (see Appendix B.3, pages 17-18,  highlighted in blue).
>
> We note that TOFU restricts model size to ≤7B because it requires using the officially released fine-tuned checkpoints as original models; therefore, larger-model experiments cannot be performed within the TOFU benchmark.
>
> | Model | Llama-3-8B |  |  | Yi-34B-Chat |  |  |
> | --- | --- | --- | --- | --- | --- | --- |
> | Metric | WMDP-Bio $\downarrow$ | WMDP-Cyber $\downarrow$ | MMLU $\uparrow$ | WMDP-Bio $\downarrow$ | WMDP-Cyber $\downarrow$ | MMLU $\uparrow$ |
> | Original | 0.472 | 0.708 | 0.631 | 0.753 | 0.497 | 0.726 |
> | RMU | 0.469 | 0.283 | 0.572 | 0.512 | 0.290 | 0.706 |
> | UUE | 0.316 | 0.229 | 0.590 | 0.452 | 0.274 | 0.710 |
>
> ### **W2: Correction of writing and formatting issues.**
>
> Thank you for pointing out the writing and formatting issues in the manuscript. We have corrected all punctuation and formatting issues, including the missing periods at line 36 and line 114. All corrections are highlighted in blue.
>
> ### **W3: Improved clarity and conciseness of the null-space construction.**
>
> Thank you for the suggestion. We have revised Section 3.2 accordingly by (1) clarifying the core condition for preserving retained knowledge (lines 150-152), (2) simplifying the explanation of the projection step (lines 167-175).
>
> ### **W4: UUE is mathematically distinct from gradient ascent and produces controlled outputs.**
>
> We address your concern as follows:
>
> 1. UUE does not produce gibberish. Experiments on TOFU benchmark show that ROUGE-L scores are 0.9+ (Table 1-2) on the retain set, indicating that output quality remains excellent. Reduced fluency on the forget set is expected for all unlearning methods, not a UUE-specific problem, since semantic coherence on the forget set is not included in unlearning goals.
> 2. UUE is mathematically distinct from gradient ascent. Unlike gradient ascent, which maximizes the likelihood of the forget set with iterative updates and is inherently unstable, our method formulates a convex quadratic minimization problem with a closed-form optimum. This ensures stable and controlled model edits rather than unconstrained gradient-ascent behavior.
> 3. Moving toward $−A_F$ has semantic grounding. $A_F$ captures the model’s semantic direction for the forget set; shifting toward $−A_F$ effectively inverts that semantic direction, analogous to steering-vector–based representation engineering.
> 4. Unlearning aims for the model to stop reproducing forgotten information while preserving its general utility. Accordingly, current unlearning benchmarks (e.g., TOFU, WMDP) evaluate performance degradation on forget data rather than the coherence or readability of the resulting outputs, since textual fluency on the forget set is not part of the unlearning objective.
>
> ### **Q1: The main experiments use PEFT rather than full-parameter fine-tuning.**
>
> Our proposed methods UUE and UUE-L are both conducted using PEFT rather than full-parameter fine-tuning.
>
> - UUE adopts an adapter-tuning paradigm, where only the inserted adapter matrices U are updated during unlearning, while the original model parameters remain frozen (see Algorithm 1).
> - UUE-L extends UUE by incorporating LoRA with adapter-tuning. During unlearning, we compute the LoRA factors Q and P via the alternating updates (Eq. (8-9)), and then apply the resulting perturbation $\Delta U = QPN$ to update the corresponding adapter.
>
> We have revised the paper to include LoRA hyperparameters used in UUE-L, including rank, alpha, and dropout (see Appendix A.2, page 16, highlighted in blue).

---

> ### Author Response · Authors · 2025-11-20
> **Response to Reviewer WnTz (Part 2)**
>
> ### **Q2: Untargeted unlearning is evaluated by utility preservation and forgetting effectiveness using standard metrics.**
>
> Evaluation for untargeted unlearning follows standard metrics:
>
> (1) Utility preservation. Unlearned model should maintain its performance on the retain set.
>
> (2) Forgetting effectiveness. For questions in the forget set, desirable responses are those that no longer reproduce the specific information memorized from the original outputs, and existing studies do not impose requirements on the coherence or readability of the generated text. Widely adopted metrics include low ROUGE-L/accuracy on forget set [1,2], large BLEU divergence on forget set before and after unlearning [3].
>
> Although untargeted unlearning differs from targeted unlearning in that it avoids explicit output constraints during the unlearning process, they share unified evaluation metrics for unlearning performance.
>
> ### **Q3: UUE is most effective when applied to middle transformer layers, as demonstrated by the layer-selection sensitivity experiment.**
>
> We select a small range of middle consecutive transformer blocks for unlearning, following prior works [4-7] showing that: (1) Early transformer layers focus on shallow linguistic patterns, while late transformer layers aggregate global semantics for final prediction, making them highly sensitive to direct editing or fine-tuning. (2) Middle layers act as the primary key–value memory responsible for storing factual associations, and editing these layers yields precise knowledge modification with minimal disruption to overall model behavior.
>
> To further support our design choice, we conducted a layer-selection sensitivity experiment on the TOFU forget01 task using Phi-1.5B. We compared editing (1) early layers (L1–6), (2) late layers (L18–23), (3) middle layers (L10–15). Results confirm that unlearning is most effective when applied to middle transformer layers, whereas early-layer edits damage overall model quality and late-layer edits fail to remove undesired knowledge.
>
> | Edited layer range | FQ ↑ | MU ↑ | ROUGE-L Forget ↓ | ROUGE-L Retain ↑ |
> | --- | --- | --- | --- | --- |
> | Early (L1–6) | 0.0289 | 0.4581 | 0.3920 | 0.6717 |
> | Middle (L10–15, main) | 0.0541 | 0.5204 | 0.3178 | 0.9228 |
> | Late (L18–23) | 0.0063 | 0.5170 | 0.6552 | 0.9201 |
>
> We have added implementation details for layer selection in Appendix A.2, and sensitivity analysis in Appendix B.5.
> ### **Q4: UUE generalizes robustly across diverse prompts and reasoning scenarios.**
>
> Our experiments are specifically designed to evaluate across different prompt formats, task types, and reasoning scenarios.
>
> (1) Generalization across heterogeneous prompt distributions. We evaluate UUE on both TOFU and WMDP benchmarks, which feature fundamentally different prompting styles. TOFU mainly consists of factoid question–answer prompts tied to synthetic biographies, whereas WMDP involves long-form hazardous-domain documents and task-specific multiple-choice queries. UUE maintains strong performance across these heterogeneous prompt distributions, demonstrating robustness under substantial input-style variation.
>
> (2) Generalization in broader reasoning scenarios. The TOFU benchmark evaluates model utility across multiple semantic domains—including fictitious authors, real authors, and world-fact knowledge. Appendix Table A1 reports detailed metrics (ROUGE-L, Probability, and Truth Ratio) for each subset, comparing UUE with all baselines. The results show that UUE consistently preserves the model’s general performance across these diverse evaluation settings, while achieving strong unlearning performance.
>
> (3) UUE’s strong generalization ability primarily comes from the null-space projection, which restricts updates to directions orthogonal to retain-set representations. This design prevents interference with broad linguistic and reasoning features of the model. The ablation in Figure 3 shows that removing the null-space constraint leads to notable drops in utility, confirming its essential role in supporting generalization.
>
> ### **Conclusion**
> We are deeply grateful for your detailed feedback, which has significantly improved our manuscript. Following your suggestions, we have made extensive revisions (all in blue): Major additions - Llama-3-8B/Yi-34B experiments (Table A4, Appendix B.3, page 17-18), Layer selection analysis (Table A6, Appendix B.5, page 19), Revised Section 3.2 (page 3-4); Technical clarifications - LoRA and layer selection hyperparameters (Appendix A.2, page 16). All formatting issues corrected.
>
> **References:**
>
> [1] TOFU: A Task of Fictitious Unlearning for LLMs
>
> [2] The WMDP Benchmark: Measuring and Reducing Malicious Use With Unlearning
>
> [3] LLM Unlearning via Loss Adjustment with Only Forget Data
>
> [4] What Does BERT Look at? An Analysis of BERT’s Attention
>
> [5] Layer-Wise Analysis of Transformer Models
>
> [6] Locating and Editing Factual Knowledge in GPT
>
> [7] Mass-Editing Memory in a Transformer

---

> ### Author Response · Authors · 2025-12-03
> **Summary of Response to Reviewer WnTz**
>
> Thank you for your thoughtful review and valuable feedback. We appreciate your **recognition of UUE’s novelty and completeness of experiments, noting that UUE enables flexible, scalable unlearning**. Your concerns and our corresponding responses are summarized as follows:
>
> (1) **Scalability to Modern LLMs.** We added **experiments on Llama-3-8B and Yi-34B-Chat (Appendix B.3)**, confirming that UUE **scales effectively to modern large LLMs**.
>
> (2) **Clarity of Method Presentation.** We **revised Sections 2–3** to resolve writing and formatting issues and improve exposition.
>
> (3) **Distinction from Gradient Ascent.** We provided theoretical and empirical evidence showing that **UUE’s closed-form null-space projection is fundamentally different from GA**, enabling **stable, semantically controlled updates** rather than unconstrained representation drift.
>
> (4) **Fine-Tuning Strategy and Hyperparameter Specification.** We clarified that all experiments use **PEFT (adapter tuning and LoRA) instead of full fine-tuning** and added complete LoRA hyperparameters (Appendix A.2).
>
> (5) **Unlearning Quality and Utility Preservation.** We explained that UUE evaluates unlearning performance using standard retain-utility and forget-effectiveness metrics (following TOFU and WMDP), maintains **high-quality outputs on retain data**.
>
> (6) **Layer Selection for Unlearning.** We theoretically and empirically justified that inserting adapters in middle transformer layers is most effective, as shown by our layer-selection sensitivity study (Appendix B.5).
>
> We regret that you did not participate in the discussion, and we believe we have fully addressed the concerns raised in your initial assessment.
>
> Best regards,
>
> Authors

---

### Official Review · Reviewer_XNzh · 2025-11-01

**Soundness:** 4
**Presentation:** 3
**Contribution:** 4
**Rating:** 6
**Confidence:** 4

**Summary:**

This paper introduces UUE, a lightweight, controllable framework for untargeted LLM unlearning that casts the problem as null-space-guided model editing to erase undesired knowledge while preserving general utility. Unlike fine-tuning-based methods that risk degrading performance due to knowledge entanglement, UUE inserts pluggable unlearning adapters after FFN layers in selected transformer blocks, freezing the backbone and updating only adapter parameters. It proposes a novel untargeted editing objective that inverts forget-set representations toward their antipodes (rather than fixed refusal outputs), ensuring convex, stable optimization without explicit targets. By projecting updates into the left null space of retain-set representations—computed efficiently via EVD on the Gram matrix—it derives a closed-form analytical solution that minimizes interference with retained knowledge. A low-rank variant, UUE-L, integrates LoRA to further reduce trainable parameters and enable flexible deployment. Extensive experiments on TOFU (fictitious author forgetting) and WMDP (hazardous knowledge removal) across Phi-1.5B, Llama2-7B, and Zephyr-7B show UUE and UUE-L achieving state-of-the-art forget quality (FQ) and model utility (MU), outperforming baselines like GA, GradDiff, IDK, and RMU in both efficacy and preservation.

**Strengths:**

- The theoretical foundation is solid: the null-space projection theorem, closed-form update derivation, and convexity proof of the untargeted objective form a coherent, mathematically rigorous pipeline that directly addresses optimization instability in prior untargeted methods.
- Experimental design is comprehensive, covering two distinct unlearning paradigms (entity-level and hazardous knowledge), multiple model families and sizes, three TOFU difficulty levels, and fine-grained metrics (ROUGE-L, FQ, MU subcomponents), with ablations validating each component’s contribution.
- Writing and structure are professional and accessible: motivation is clearly framed with intuitive figures (targeted vs. untargeted), method sections build logically from adapter design to closed-form solution to LoRA extension, and results are presented in well-organized tables with statistical significance.

**Weaknesses:**

Model scale evaluation is limited to ≤7B parameters; claims of scalability and lightweight deployment would be more convincing with results on 13B+ or 70B-class models, where memory and compute bottlenecks are more pronounced.

**Questions:**

- The eigenvalue threshold for null-space approximation is manually tuned—have you experimented with data-driven selection (e.g., cumulative variance) or observed failure modes when γ is misspecified?
- Since UUE generates diverse, non-refusal responses on forget queries (sometimes creative or hallucinated), how do you ensure this doesn’t violate safety constraints in regulated domains like biosecurity?

---

> ### Author Response · Authors · 2025-11-20
> **Response to Reviewer XNzh**
>
> We appreciate your thoughtful feedback and your recognition of our work. We have provided detailed clarifications for all the points you raised.
>
> ### **W1: UUE scales effectively to 34B-parameter models.**
>
> Thank you for the suggestion. We additionally evaluate UUE on larger modern LLMs—Llama-3-8B-Instruct and Yi-34B-Chat—using the WMDP benchmark. The results show that UUE effectively reduces hazardous knowledge while maintaining MMLU performance close to the original models, and consistently outperforms RMU. These results confirm that UUE remains scalable to larger LLMs. We supplemented the corresponding experiments and analyses in the revised manuscript (see Appendix B.3, pages 17-18,  highlighted in blue).
>
> We note that TOFU restricts model size to ≤7B because it requires using the officially released fine-tuned checkpoints as original models; therefore, experiments using larger models cannot be performed on the TOFU benchmark.
>
> | Model | Llama-3-8B |  |  | Yi-34B-Chat |  |  |
> | --- | --- | --- | --- | --- | --- | --- |
> | Metric | WMDP-Bio $\downarrow$ | WMDP-Cyber $\downarrow$ | MMLU $\uparrow$ | WMDP-Bio $\downarrow$ | WMDP-Cyber $\downarrow$ | MMLU $\uparrow$ |
> | Original | 0.472 | 0.708 | 0.631 | 0.753 | 0.497 | 0.726 |
> | RMU | 0.469 | 0.283 | 0.572 | 0.512 | 0.290 | 0.706 |
> | UUE | 0.316 | 0.229 | 0.590 | 0.452 | 0.274 | 0.710 |
>
>
> ### **Q1: $\gamma$ =0.01 is robust across different settings.**
>
> Thank you for the insightful comment. We choose a small fixed threshold ($\gamma$ = 0.01) as a conservative and stable criterion for identifying near-zero eigenvalues. To evaluate its robustness, we conducted an additional experiment on the TOFU forget01 task (Phi-1.5B) using fixed and cumulative variance-based threshold selection rules:
>
> 1. Fixed absolute threshold: $\gamma \in$ {0.001,0.01,0.1}.
> 2. Cumulative explained variance (CEV): selecting the smallest set of eigenvalues that satisfies $\frac{v_j}{\sum_{i=1}^d v_i} < \gamma,\gamma \in $ {0.001,0.01}.
>
> Across all settings, UUE remains stable for small–moderate thresholds and only degrades when $\gamma$ becomes unrealistically large, as this removes meaningful retained directions and makes the null-space projection overly aggressive. Moreover, while CEV offers a principled alternative, our experiments indicate no measurable advantage in this setting. This confirms that a small fixed threshold is a reliable and robust strategy.
>
> We supplemented the corresponding experiments and analyses in the revised manuscript (see Appendix B.4, pages 17-19, highlighted in blue).
>
> | Thresholding rule | $\gamma$ | FQ ↑ | MU ↑ | ROUGE-L Forget ↓ | ROUGE-L Retain ↑ |
> | --- | --- | --- | --- | --- | --- |
> | Fixed  | 0.001 | 0.0423 | 0.5211 | 0.3321 | 0.9225 |
> | Fixed | 0.01 | 0.0541 | 0.5204 | 0.3178 | 0.9228 |
> | Fixed | 0.1 | 0.0540 | 0.5039 | 0.3154 | 0.9081 |
> | CEV | 0.001 | 0.0530 | 0.5199 | 0.3205 | 0.9213 |
> | CEV | 0.01 | 0.0544 | 0.5195 | 0.3150 | 0.9220 |
>
> ### **Q2: UUE does not introduce new safety risks.**
>
> Thank you for raising this important point. UUE is designed to avoid adding harmful behaviors and not introduce new risks, which we demonstrate from the following perspectives:
>
> (1) From a theoretical aspect, UUE’s null-space projection mechanism constrains parameter updates to directions orthogonal to retained knowledge. Meanwhile, UUE does not introduce any new harmful knowledge during unlearning. These ensure that the resulting outputs are not completely uncontrolled and do not produce new unsafe information.
>
> (2) From an empirical aspect, experiments on the WMDP benchmark confirm that UUE consistently reduces hazardous model behavior as intended. Performance on hazardous queries drops substantially, confirming that the method suppresses unsafe content rather than generating new risks.
>
> (3) For practical applications, it is not sufficient to rely solely on model-side unlearning. As with any unlearning method, safety-critical applications (e.g., biosecurity, cybersecurity) should combine unlearning with standard safety techniques such as input and output filtering, logging, monitoring, etc. The primary goal of our paper is to enhance the effectiveness of unlearning rather than to offer a comprehensive LLM safety benchmark.
>
> ### **Conclusion**
> Once again, we sincerely appreciate your thoughtful feedback. To address your suggestions, we have made the following revisions: (1) Added Llama-3-8B/Yi-34B experiments (Table A4, Appendix B.3, page 17-18, highlighted in blue), (2) Added threshold sensitivity analysis (Table A5, Appendix B.4, page 17-19, highlighted in blue). We believe these revisions substantially strengthen the paper.

---

> ### Author Response · Authors · 2025-12-03
>
> Thank you for your **positive assessment** of our work and for **highlighting the writing, theoretical and experimental advantages strengths of the paper**. We believe the revisions comprehensively address the concerns you raised, and again we sincerely appreciate your constructive initial feedback.
>
> Best regards,
>
> Authors

---

### Author Response · Authors · 2025-12-03
**Summary of Discussion**

Dear PCs, SACs, ACs, and Reviewers,

We sincerely appreciate your time and effort devoted to reviewing and discussing our work. **We appreciate the positive comments on UUE’s theoretical soundness, novelty, and comprehensive experimental evaluation.** We have carefully addressed all comments and revised the manuscript accordingly. All modifications are **highlighted in blue (additional experiments) and orange (contextual and related-work enhancements)**.

## Summary of Reviews and Responses

1. **Scalability to Modern Large LLMs.** Reviewers XNzh, WnTz and 43Hv acknowledge **UUE’s lightweight design and comprehensive evaluations**. We add **experiments on Llama-3-8B and Yi-34B-Chat (Appendix B.3, blue)**, confirming that **UUE remains effective and lightweight** on modern, larger LLMs, which can solve the concern raised by reviewers XNzh and WnTz.
2. **Methodological Justification.** Reviewers XNzh and 43Hv recognize UUE’s novelty and solid theoretical foundation. In response to reviewer WnTz, we **clarify that UUE is fundamentally distinct from gradient ascent**—it is a **closed-form, null-space–guided optimization framework** that enables **stable and controllable unlearning updates**, rather than unconstrained representation drift. We further **add a layer-selection sensitivity analysis (Appendix B.5, blue)**, providing **theoretical and empirical evidence** that middle transformer layers are most suitable for effective unlearning, which can solve the concern raised by WnTz and 43Hv.
3. **Quality and Safety of Post-Unlearning Outputs.** Reviewers WnTz and  43Hv recognize UUE’s strong unlearning performance. In response to reviewers XNzh and WnTz’s concerns regarding the quality and potential safety risks of model outputs post-unlearning, we clarified—**supported by extensive experimental results**—that UUE **reduces harmful content generation**, and maintains **high-quality outputs on retain data**.
4. **Improved Writing and Conceptual Clarity.** Following reviewers’ suggestions, we enhanced the manuscript’s clarity by **strengthening the motivation for method design, explaining the retain-set construction, and adding supporting citations** **(Sec. 1–3, orange)**.

## Summary of Reviewer Feedback

**Reviewer 43Hv** confirmed that our responses **fully resolve concerns raised and upgraded their score to 6 (at 22 Nov 2025, 20:19)**. **Reviewer XNzh and WnTz did not participate further**, though we believe the provided clarifications address all raised issues.

The final score is **6** (XNzh), **6** (43Hv), **4** (WnTz).

## Summary of Contribution

We summarize our contributions as follows:

- **Novel untargeted unlearning framework.** We reformulate LLM unlearning as **null-space–guided untargeted model editing** and introduce **UUE, a lightweight and controllable untargeted unlearning framework** that overcomes the core limitations of prior methods—specifically, over-forgetting, costly fine-tuning, and reliance on ad-hoc target outputs.
- **Methodological innovation.** UUE employs pluggable unlearning adapters and closed-form analytical updates to constrain unlearning updates within the null space of retained knowledge, ensuring stable optimization and minimal disruption to model utility.
- **Empirical effectiveness and scalability.** Extensive experiments on TOFU and WMDP across diverse LLM architectures (up to 34B parameters) confirm that UUE and UUE-L achieve state-of-the-art unlearning performance while substantially reducing computational overhead.

Best regards,

Authors

---

### Meta-Review · Area_Chair_uTmt · 2025-12-11

**Summary:**

This paper proposed lightweight, theoretically tidy null-space editing with competitive performance and some larger-model evidence. However, general validity remains limited (few modern models, no large-scale experiments), comparisons to concurrent methods are incomplete, and the practical value of untargeted outputs on forget queries (coherence/usefulness) is under-substantiated. Safety assurances rely on proxy benchmarks rather than end-to-end audits. Given these remaining gaps despite a strong rebuttal, I recommend reject.

**Reviewer Concerns:**

**XNzh** Concerns: limited scale and risks to safety/quality; clarification on null-space thresholding. Addressed: added Llama-3-8B and Yi-34B results; safety rationale; threshold robustness study. Outstanding: external-validity remains modest (few large models, no TOFU at large scale); safety evidence rests on benchmark proxies rather than task audits.

**WnTz** Concerns: outdated model choices; writing/organization; whether method reduces to gradient ascent; semantic quality on forget set; layer selection; PEFT specifics and metrics. Addressed: modern-model results; edits to exposition; theoretical distinction from GA; PEFT/LoRA hyperparams; layer-sensitivity; metric clarification. Outstanding: persistent doubts about coherence/utility of untargeted outputs on forget queries and breadth of real-world validation.

**43Hv** Concerns: missing references and motivations (untargeted choice, FFN placement, retain-set design, embedding-space edits), comparative baselines, and layer selection. Addressed: added references, comparisons (partial), and sensitivity studies. Outstanding: comparisons not yet comprehensive across all settings; qualitative post-unlearning behavior only lightly probed.

**Reviewer Scores:**

**XNzh** Likely unchanged at 6 (concerns partially addressed).

**WnTz** Likely unchanged at 4 (core validity/coherence doubts remain).

**43Hv** Raised to 6 given satisfaction in discussion (which cannot be counted due to the leak and possible collusion).

> **Why Reverting Back?** We made the decision to revert the discussion back to prior to the discussion period because the leak occurred as early as November 11th (before the discussion). We consequently have to assume that collusion could have occurred at any point during the discussion phase. After extensive discussion, we found reverting the scores to the beginning of the discussion phase to be the fairest course of action for all authors.

---

### Decision · Program_Chairs · 2026-01-26

Reject